

# Post-event Field Survey of 28 September 2018 Sulawesi Earthquake and Tsunami

Wahyu Widiyanto[1,2], Purwanto B. Santoso[2], Shih-Chun Hsiao[1], Rudy T. Imananta[3]

[1]Department of Hydraulic and Ocean Engineering, National Cheng Kung University, Tainan, 701, Taiwan
[2]Department of Civil Engineering, Universitas Jenderal Soedirman, Purwokerto, 53122, Indonesia
[3]Meteorological, Climatological and Geophysical Agency (BMKG), Jakarta, 10720, Indonesia

*Correspondence to*: Shih-Chun Hsiao (schsiao@mail.ncku.edu.tw)

**Abstract.** An earthquake with a magnitude of $M_W = 7.5$ that occurred in Sulawesi, Indonesia on September 28, 2018, triggered liquefaction and tsunamis that caused severe damage and many casualties. This paper reports the results of a post-
10 tsunami field survey conducted by a team with members from Indonesia and Taiwan that began 13 days after the earthquake. The main purpose of this survey was to measure the runup of tsunami waves and inundation and observe the damage caused by the tsunami. Measurements were made in 18 selected sites, most in Palu Bay. The survey results show that the runup height ranged from 2 to 10 m and that the inundation distance was between 80 and 510 m. The highest runup (10.5 m) was recorded in Tondo, a complex that has many boarding houses near a university. The longest inundation distance (511 m) was
15 found in Layana, a marketplace. The arrival times of the tsunami waves were quite short and different for each site, typically about 3-8 minutes from the time of the earthquake event. The characteristics of the damage to buildings, facilities, and structures are also summarized. Several indicators of underwater landslides are described. The survey results can be used for the calibration and validation of hydrodynamic models for tsunamis. They can also be used for regional reconstruction, mitigation, planning, and development.

**1 Introduction**

On Friday, September 28, 2018, at 18:02:44 Central Indonesia Time (UTC + 8), Palu Bay was hit by a strong earthquake with magnitude $M_W = 7.5$. The epicenter was located at 0.18°N 119.85°E at a depth of 11 km and 27 km northeast of Donggala City. The major phenomena following the earthquake were liquefaction and tsunamis. As of October 21, 2018, as many as 2,113 people were killed, 1,309 missing, and 4,612 injured. In addition, 66,238 houses were damaged. The source
of the earthquake was the shift of the Palu-Koro strike-slip fault, one of the most active structures around Sulawesi. After the earthquake, a series of tsunami waves hit Palu City and Donggala Regency. Low-amplitude tsunami waves were also detected in Mamuju, a city overlooking the Makassar Strait and outside Palu Bay. The tsunami hit the coast, leveled houses, washed away various objects and destroyed the coastal area of Palu Bay, Central Sulawesi Province.





Within the territory controlled by Indonesian authorities, the 2018 Sulawesi Tsunami was the most devastating since the 2004 Indian Ocean Tsunami. There were 8 tsunami events after the December 26, 2004 Indian Ocean Tsunami, namely 2005 Nias ($M_W$ = 8.6;1,314 victims), 2006 Buru Island ($M_W$ = 6.7; 4 victims), 2006 Java ($M_W$ = 7.7; 668 victims), 2007 Bengkulu ($M_W$ = 8.4; 23 victims), 2009 Manokwari ($M_W$ = 7.6; 4 victims), 2009 Tasikmalaya ($M_W$ = 7.3;79 victims), 2010 Mentawai

($M_W$ = 7.8; 413 victims), and 2018 Sulawesi ($M_W$ = 7.5;2,113 victims). These tsunami events are distributed in tsunami zones that cover all parts of Indonesia except Kalimantan island. Refering to the tsunami catalog and zones in Indonesia (Latief et al., 2000), the 2018 tsunami was on the border between zone D, which includes the Makassar Strait, and zone E, which includes the Maluku Sea. Zones D and E accounted for 9% and 31%, respectively, of the total tsunami events in Indonesia between 1600 and 2000. The Palu-Koro fault where the 2018 tsunami occurred is a very active source of

earthquakes and tsunamis in zones D and E.

The Palu-Koro fault, which divides Sulawesi into two parts, has quite active tectonic activity. The movement of rock formations is 35-44 mm/year (Bellier et al., 2001). This is the second most active fault in Indonesia after the Yapen fault in Papua. The Sulawesi region has a long history of earthquakes and tsunamis (Prasetya et al., 2001). On January 30, 1930, an earthquake occurred on the West Coast of Donggala that caused a tsunami with a height of 8-10 m, 200 deaths, 790 houses

damaged, and the flooding of all villages on the west coast of Donggala. On January 1, 1996, an earthquake in the Makassar Strait caused a tsunami that swept the west coast of Donggala and Toli-Toli Districts. In the same year, an earthquake occurred in Bangkir Village, Tonggolobibi, and Donggala, causing a 3.4-m-high tsunami that carried sea water 300 m inland, 9 people were killed and buildings in Bangkir, Tonggolobibi, and Donggala villages were badly damaged. On October 11, 1998, another earthquake occurred in Donggala, severely damaging hundreds of buildings. In 2005 and 2008, earthquakes

also occurred, but did not cause many casualties. The most recent earthquake occurred in Sigi Regency and Parigi Moutong Regency in August 2012, which left 8 people dead.

The disaster area of the September 2018 tsunami includes Palu Bay, a bay on Sulawesi island, and Central Sulawesi Province. This bay has a length of 30 km, a width of 7 km, and a maximum depth of 700 m. Although the epicenter was at the outer boundary of Palu Bay, the most severe damage suffered in Palu City was at the end of the bay, about 70 km from

25 the epicenter. Palu City, the capital of Central Sulawesi Province, has a population of 379,782 (BPS-Statistics of Palu Municipality, 2018). The most victims came from this city. In addition to Palu City, the disaster area also included Donggala Regency, with a population of 299,174 (BPS-Statistics of Donggala Regency, 2018), and Sigi District, with a population of 234,588 (BPS-Statistics of Sigi Regency, 2018). Sigi Regency did not suffer damage from the tsunami, but large-scale liquefaction led to a significant number of deaths and disappearances in this area.

This disaster in Central Sulawesi has astonished earthquake and tsunami experts and researchers. Geologically, the area of the earthquake includes the Palu-Koro fault, an active strike-slip fault in Indonesia. For this type of fault, the plates move horizontally and thus do not usually cause enough vertical deformation to trigger a huge tsunami. However, the tsunami in Central Sulawesi destroyed property and caused many casualties. Several hypotheses have been proposed. One is that there was an underwater landslide, which was the driving force for the tsunami wave. To answer such questions and other





questions that arise from a tsunami event, field surveys play an important role. For tsunamis, post-incident surveys are often carried out. Major tsunamis such as the 2004 Indian Ocean Tsunami and the 2011 Tohoku Tsunami require many teams to conduct surveys. Field surveys for the Indian Ocean Tsunami have been reported by Borrero (2005), Fritz and Borrero (2006), and Goff et al. (2006). Field surveys for the 2011 Tohoku Tsunami have been reported by Mori et al. (2011), Mikami

et al. (2012), Liu et al. (2013), and Suppasri et al. (2013).

The focus of post-tsunami surveys depends on the data of interest (e.g., hydrodynamic, geological, geophysical, environmental, ecological, social, or economic). The field survey reported in the present study focuses on hydrodynamic data that includes measurements of runup height and inundation distance. Tsunami water depth on land was also measured in some areas. Observation of damage was also conducted. The data can be used for the simulation of tsunamis caused by plate

movements or underwater landslides. For instance, Lynett et.al (2003) employed the field survey data of the 1998 Papua New Guinea tsunami as validation for numerical model, i.e. Boussinesq model and a nonlinear shallow water wave model. More broadly, these data can be used for disaster mitigation and rebuilding the affected areas of the 2018 Sulawesi Tsunami.

## 2 Survey Details

A team from National Cheng Kung University, Taiwan, and Universitas Jenderal Soedirman, Indonesia, arrived at Sis Aljufri

Airport in Palu City at 06:00 Central Indonesia Time on October 11, 2018, thirteen days after the tsunami event. Studies have shown that surveys can be carried out successfully within two to three weeks of an event (Synolakis and Okal, 2005). Starting from the afternoon of October 11, a field survey was conducted until October 19 evening, for a survey period of 9 days. The emergency response period for the disaster area was determined by the Indonesian government to be one month (September 28 to October 26, 2018). The victim evacuation period was two weeks (September 28 to October 12). This

means that the survey was conducted in the emergency response stage, one day before the victim evacuation period ended.

Our survey covered the following activities: 1) gathering information about disaster-affected locations, collecting videos and photographs of tsunami events from the news, websites, social media, and personal collections of residents that had experienced the disaster; 2) tracing the road along the coast in Palu Bay to get an overview of the affected area; 3) choosing a measurement site that can represent the impact of the tsunami; 4) looking for evidence of runup boundaries, inundation

limits, and tsunami water level elevation from the subgrade; 5) measuring the profile of the beach hit by the tsunami; 6) observing and documenting specific damage and phenomena; and 7) interviewing eyewitnesses.

Because many people have smartphones, documentation in the form of photographs and videos is abundant. Such documentation was collected from the internet. Unfortunately, many people with valuable documentation did not upload it to the internet. Therefore, our team searched for video recordings and photographs made by local residents while conducting

the measurement survey. We also recorded videos and took photographs.

The disaster location was located in Palu Bay. The survey area covers the entire coastal area in the bay, which falls under the authority of Central Sulawesi Province. The coastline in the bay is around 70 km. The road connecting the provinces on



Sulawesi island, called the Trans Sulawesi Road, is mostly parallel to the coastline of the bay. Our team traced the road from Donggala City to Sirenja Village, which is the limit of the tsunami disaster in Palu Bay. Tracing the Trans Sulawesi Road to see an overview of the impact of the tsunami is possible because the road is mostly located 50 to 200 m from the coastline, so the coastline can almost always be seen from the road. We operated a camera on a moving vehicle to record the situation

around the beach area. This method is similar to that used by Google Street View®, but we used simpler equipment.

We chose 18 locations to measure (Fig. 1). These locations were used to represent runup and inundation data in Palu Bay. At each site, the beach profile was measured using 1 to 4 transects or cross sections, for a total of 28 cross sections. Site selection was based on consideration of the level of damage, significance of runup height and length of inundation, as well as resources and time. Many locations with steep cliffs and tsunami trails were not easily visible. We did not take

measurements in such locations. Likewise, we did not measure places not significantly affected by the tsunamis. The measurement times and locations of the 18 sites are shown in Table 1. The table gives the runup height and inundation distance, which are explained in section 3.

Finding evidence of runup heights, boundaries of inundation, and elevation of tsunami water levels is challenging. Some

detective work is often necessary. October is the beginning of the rainy season in Indonesia, including Sulawesi. Palu City is located near the equator, as shown by latitudes in Table 1. It is one of the driest areas in Indonesia, with rainfall recorded at the Mutiara Meteorology Station in 2017 of 774.3 mm. Since the earthquake incident until the date of our team's return, it rained four times, three of which occurred during our survey period, with a duration of less than 2 hours and with low to moderate intensity. Fortunately, from the point of view of conducting a survey, surface runoff due to rain seems insignificant

and does not erase the tsunami footprint. Our team collected hundreds of traces and water marks left by the tsunami. The tracks were in the form of: a) debris lines, b) debris left on trees, c) broken branches, d) sand trapped in buildings, e) damaged building elements, and f ) brown leaves (submerged in salt water during tsunami event). Figure 2 shows some evidence of runup and inundation traces.

In addition to physical evidence that could be seen and documented in the field, eyewitnesses are important because they

directly experienced the earthquake and tsunami. Very often interviews provide unique information that cannot be obtained by any other means and are therefore much more than an auxiliary tool (Maramai and Tinti, 1997). Our team interviewed 56 people throughout the disaster area. Some of the interviews were recorded in video form so the testimonies can be heard again. We got some important information from the interviews, such as the arrival time of the tsunami, the number of waves coming in, the boundaries of runup and water level, the situation in the area before and after the tsunami, the magnitude of

earthquake shocks, and how to save themselves from the tsunami. We were told that there were 3 tsunami waves. The first wave had the smallest amplitude. Then, two waves followed it. The first wave was a warning so many people went away from the coastline immediately. Without this first low-amplitude wave, there may have been more casualties.

After the physical evidence and/or witnesses confirmed the position of the entry of tsunami water inland, measurements were carried out using conventional measurement instruments. Our team operated several laser and optical instruments for



terrestrial surveys. The instruments were a total station, a water pass, a prism, a handheld GPS device, laser distance meter, and some assistance tools. These tools were used to measure height differences and the distance from a point and position.

## 3 Runup and Inundation Observation

Runup is the maximum ground elevation wetted by the tsunami on a sloping shoreline. In the simplest case, the runup value
is recorded at the maximum inundation distance, the horizontal distance flooded by the wave (IOC Manuals and Guides No. 37, 2014). Because of the difficulty of using a benchmark on land, we measured runup height to the local sea level at the time of measurement and then corrected it to above sea level on average using astronomical pairs of data collected from tidal stations in Mamuju and Pantoloan. Mamuju is located outside the bay, on the west coast of Sulawesi overlooking the Makassar Strait. Pantoloan is located inside Palu Bay. The measurement results are shown in Table 1 and Figs. 3-5.
From Table 1 and Fig. 3, the farthest inundation distance from coastline to tsunami border on land is at the Layana site. This site is a trade complex that supports the economic activities of Palu City in particular and Central Sulawesi Province in general. The buildings damaged at this site functioned as shops, warehouses, and corporate offices. The items scattered in the tsunami were predominantly manufactured goods of economic value. This site is thus monitored closely by security forces to prevent looting. The topography of this site is relatively flat with a slope of 0.013 (1.3%). Because of this sloping
topography, the tsunami wave reached as far as 511 m inland. The farthest point reached in this area varies greatly because many buildings have long and wide walls that stemmed the tsunami flow further inland.

The highest point of the tsunami creeping was found at the Tondo site (Fig. 5). This area has many private boarding houses for students of the University of Tadulako, a state university in the city of Palu. The topography of this area is relatively steep with a slope of 0.06 (6%). Evidence of tsunami water rise was in the form of debris on top of buildings, truncated
building elements, collapsed walls, trash carried away, and fixed debris. Eyewitnesses also showed the highest places of tsunami water in this area. A total of 4 cross sections of these coast were measured by our team. The measured runup heights were 12.20, 9.44, 11.61, and 9.97 m, respectively, as shown in Table 1. The runup height of 12.2 m (10.5 m after correction based on astronomical tides) is the highest at the Palu disaster area site. This area was flattened by tsunami (Fig. 6a), buildings collapse. This area is very crowded. Most students were in their boarding houses during the earthquake because it
occurred after working hours. Surprisingly, fewer than 10 deaths were recorded. This is likely due to most of the young residents being quite nimble and able to save themselves when the tsunami arrived. This is an important observation for future mitigation efforts.

Records are also shown for sites 7, 8, and 9 (Lere, Besusu, and Talise, respectively). The area of these sites is at the end of the bay, has a sloping topography, the highest population, the most fatalities, and the worst damage. The runups were not
very high (less than 2 m), but the inundation distance was relatively long (270-290 m). The density of buildings in this area seems to have prevented the tsunami from reaching further inland. Runup data from this survey can be utilized to support runup modeling especially for densely populated areas as promoted by Muhari et al. (2011).





## 4 Damage due to Tsunami

The Meteorological, Climatological and Geophysical Agency (BMKG) stated that the Sulawesi 2018 earthquake had a scale of Modified Mercalli Intensity (MMI) VIII (severe). Scale VIII MMI is characterized by minor damage to specially designed structures; major damage to ordinary large buildings with partial collapse; major damage to poorly constructed structures;

the fall of chimneys, piles of factories, columns, monuments, and walls; and heavy furniture being turned over. Our team documented much of the scale VIII MMI evidence that could be seen in disaster areas. Some areas even had scale IX MMI. Damage to buildings and structures at the disaster site can be divided into 3 types, namely damage due to earthquakes, liquefaction, and tsunamis. Damage caused by earthquakes is characterized by horizontal collapse, cracking, and fracture structures. Damage due to liquefaction can be characterized by objects and buildings being turned over, rotated, gone, sunk

in water, or sunk in mud. Damage due to tsunamis is characterized by objects, buildings, or structures being washed away from the shoreline by a water current.

Liquefaction was a significant event in the disaster in Central Sulawesi. There are 3 locations of liquefaction with a large area and damage, namely Petobo and Balaroa in Palu City and Jono Oge in Sigi Regency. The corresponding areas of liquefaction were180 hectares, 48 hectares, and 202 hectares, and the numbers of houses damaged or destroyed were 2050,

1045, and 366, respectively. The land dropped by 3 m and rose by 2 m. Before the September 2018 incident, an investigation of potential liquefaction in Palu City and its surroundings was carried out by Widyaningrum (2012).

A reinforced concrete bridge on Cumi-cumi Road, Palu City (**Error! Reference source not found.**b), gives a clue regarding the tsunami's strength. This bridge is made of reinforced concrete with a bridge span of 5.0 m and a width of 19.1 m. This

bridge has a relatively small span because it passes over a sewer from the city of Palu, not a large river. The sewer has a width of 4.1 m and a depth of 1.6 m. The width of the bridge is relatively large (two lanes). The bridge has14 beam girders with dimensions of 0.25 m × 0.30 m with a girder distance of 1.35 m. The bridge has simple support. The bridge plate has a thickness of 0.20 m. Based on these dimensions, the mass of the bridge was estimated to be around 38 tons. A large and perpendicular direction of the tsunami's velocity shifted the bridge by as far as 9.7 m without damage. The surface area of

the bridge perpendicular to the direction of arrival of the tsunami is 3.4 m$^2$, including the area of the projected bridge fence. The bridge was estimated to have been submerged by tsunami water as deep as 2.5-4.0 m based on the tsunami marks around it. Debris caught in the bridge fence (Fig. 6b) was evidence of the tsunami water soaking the bridge. The shift stopped because the bridge body was stuck in the wall of a building. We can investigate this case with the help of Google Earth, as shown in Fig. 7, where Figs. 7a and 7b show satellite images taken on September 26, 2017, and October 2, 2018,

respectively. As shown, the asphalt layer of the road was broken and the bridge over the sewer channel was shifted away from the coast by the tsunami. The position of this bridge is at the end of Palu Bay (-0.88123°S 119.83907°E). This phenomenon can be further analyzed to determine the tsunami force.



## 5 Additional Relevant Information

Several types of data obtained directly from sites or collected from agencies may serve as significant information for further analysis.

### 5.1 Aftershock

The aftershock data were collected from the Meteorological, Climatological and Geophysical Agency (BMKG). Figure 8 shows the distribution of aftershocks updated on October 18, 2018, at 09:00 West Indonesia Time. It can be seen that there were 561 earthquakes events for 20 days after the main earthquake with magnitudes of $M_W$ 2.5-6. The frequency of the earthquakes decreased over time, 21 of the earthquakes could be felt by people. Among them, on the day of event, September 28, 2018, from main earthquake time to 06:21 p.m. there were aftershocks with magnitudes of M6.3, M6.2, M6.2,

M4.7, M5.6, M5.0, and M6.1, respectively.

### 5.2 Tide

The tidal station closest to the disaster site is Pantoloan Tidal Station. This station is located inside Palu Bay, on a pier in Pantoloan Port and operated by the Agency of Geospatial Information. When the earthquake and tsunami occurred, the recording equipment was not damaged but the data transfer stopped because the communication network was interrupted.

Figure 9 shows the water level recorded when the tsunami arrived. The maximum low tide (6.74 m) was at 18:08 local time and the maximum tide (10.55 m) was at 18:10 local time. This means that the tsunami wave height recorded at the station was 3.8 m. This wave height can be seen in Fig. 10, which is from the same source as that for Fig. 9. In addition, the first tsunami wave arrived at 18:07, with the wave trough at 18:08 and the wave crest at 18:10 local time (UTC+8).

### 5.3 Tsunami Arrival Time

The time of arrival of a tsunami wave is one of the main parameters calculated in tsunami modeling. The time needed for the tsunami wave to propagate from earthquake source location to the coast is defined by the estimated time of arrival (ETA) (Strunz et al., 2011). It is important related to early warning system. Based on videos on social media, internet, and television, as well as eyewitnesses, more than one tsunami wave hit the beaches in Palu Bay. Most witnesses stated that three tsunami waves had arrived. The first was less than 1-m high. The second and third waves were much higher, and were

quantified by measurements in this survey. The number of tsunami waves and their height order were similar to the 17 July 2006 Tsunami in Java. That event also had three tsunami waves which the first one was of little magnitude and was followed by the second wave which was the highest one (Lavigne et al., 2007). Eyewitnesses did not give an exact arrival time of the tsunami wave on the beach. Generally, they referred to prayer times as a guide. Indonesia is majority Muslim. The time of the earthquake and tsunami is close to one of the Muslim worship times in the afternoon, which coincides with a sunset



called "maghrib" prayer. The prayer schedule circulated by the Ministry of Religion of the Republic of Indonesia for the area of Palu City and Donggala Regency indicates that the starting time of "maghrib" prayer period on September 28, 2018, was 17:58 local time. Normally, there are two call sounded from a mosque as starting time sign for praying. The first call is called "adzan" and the second call is called "iqamah". The period between the two call is 10 minutes. Some news, videos,

and witnesses show that the tsunami came when people were preparing to pray, between "adzan" and "iqamah". The $M_W = 7.5$ earthquake occurred at 18:04. This shows that the tsunami waves came less than 10 minutes after the earthquake or between 18:05 and 18:15 local time, different for each site in the disaster area. These results can be used to verify the arrival time estimated by models using tidal data. The important thing from the September 2018 event is that the tsunami arrival time was very short.

**5.4 Surface-earth Landslide**

Our team found 16 locations for earth-surface landslides. Their locations are summarized in Table 3. These locations are in hilly areas, where the slope is 60°-80°. Locations 1 and 2 are in Oti Village, locations 3 and 4 are in Batusuya Village, and locations 5 to 16 are in Sindue Tombusabora sub-district. The landslides were caused by a series of earthquakes that occurred from 13:00 local time to the largest earthquake (magnitude: 7.5), which triggered the tsunami. Our team only

recorded the coordinates of the locations of the surface landslides, not the landslide volumes. In addition, our team also only inventoried landslides that occurred along Jalan Toli-toli to Palu. Further investigation on geology or geotechnics may be needed to find a relationship between surface-earth landslides and underwater landslides. The detection of submarine landslidesis being conducted by various agencies, including the Indonesian Navy on board the KRI Spica 934 and the Technology Assessment and Application Agency (BPPT) with the Baruna Jaya 1 survey ship.

**5.5 Underwater Landslide**

A strike-slip fault earthquake with magnitude $M_W = 7.5$ should not have produced the large observed tsunami. The 2018 Sulawesi tsunami might have been generated by a mechanism other than the strike-slip earthquake. This suspected source should be located near the impacted areas with a relatively narrow width (Muhari et al., 2018). It is possible that a large submarine landslide contributed to and intensified the Sulawesi tsunami. The southern part of the Palu Bay, around

latitude-0.82°S, is the most likely location of a potential landslide (Heidarzadeh et al., 2018). Sassa and Takagawa (2019) estimated that less than 20% of the Sulawesi tsunami height was related to tectonic processes, and that the majority was caused by coastal and submarine landslides, as characterized by liquefied gravity flows. On the contrary, by using coupled earthquake-tsunami physical-base model, Ulrih et al. (2019) resume that a source related to earthquake displacement in the strike-slip system is probable and that landsliding may not have been primary source of the tsunami in Sulawesi 2018 event.

So far, source of tsunami in Sulawesi is still in discussion.





Some evidence from videos and eyewitnesses indicates that landslides or coastal land subsidence occurred inside Palu Bay area. Therefore, the Indonesian Navy deployed the KRI Spica Ship, a new survey vessel equipped with a multibeam echosounder, to record underwater data on October 15, 2018, after the tsunami. The Agency for the Assessment and Application of Technology (BPPT) sent a Baruna Jaya 1 survey ship to conduct a bathymetry survey of Palu Bay after the

tsunami. Although the focus of our team was to measure runup and inundation, we also obtained evidence that might be related to underwater landslides. Total collapses and flows of coastal land due to liquefaction occurred in at least nine locations (Sassa and Takagawa, 2019). We found two of them with indicators of land subsidence on the coast. The two locations are around the river mouth in Donggala City (Figs. 11 and 12) and around the river mouth of Lero Village (Figs. 13 and 14).

**6 Conclusions**

This study reported the results of a post-tsunami field survey conducted after the 2018 Sulawesi Tsunami. The results show that the runup heights ranged from 2 to10 m and the inundation distances were 80 to 500 m. The highest runup (10.5 m) was in the Tondo area, which has a steep slope coast. The farthest inundation (503 m) was in the Layana area, which has a flat topography. There were three main tsunami waves that reached the beach. The first wave was relatively low. Almost

all beaches in Palu Bay were hit by tsunami waves. The arrival time of waves varied from 3 to 10 minutes after the $M_W = 7.5$ main earthquake event (between 18:05 and 18:15 local time). The worst damage was at the end of Palu Bay. The results of this field survey can be used for the calibration and validation of hydrodynamic models for tsunamis.

The wave that hit the beach was quite high. A reinforced concrete bridge weighing 38 tons at the end of the Palu bay was shifted by 9.7 m without damage. This indicates that the tsunami wave in Palu has great momentum. The tsunami force

needs to be further analyzed to provide valuable data for reconstruction. Land subsidence in Donggala City (10,068 $m^2$) and Lero Village (22,971 $m^2$) gives evidence for underwater landslides. However, it does not mean there were only two underwater landslides; more landslide locations may be found in the disaster area. This event is motivation for the development of a tsunami model that is capable of simulating tsunamis generated by consecutive earthquake and landslide events, or simultaneous landslide events. Furthermore, landslides should be included in probabilistic tsunami hazard

assessment, as done for Indonesia by Horspool et al. (2014). The data and analysis from this survey and those from other teams will lead to a comprehensive and complete understanding of the September 2018 Sulawesi Tsunami.

*Data availability*. All photos were taken by author's team. Earthquake after shock graphics was from Meteorological Climatological, and Geophysical Agency (BMKG). Tide data was obtained from Geospatial Information Agency (BIG).

*Author contribution*. All authors contributed to the preparation of this paper.

*Competing interest*. The authors declare that they have no confllict of interest.



*Acknowledgement*

This research was financially supported by the Ministry of Science and Technology, Taiwan, under grants MOST 107-2221-E-006-080 and Water Resources Agency, MOEA, under grants MOEAWRA 1080412. Thanks to Universitas Jenderal Soedirman, Indonesia, especially the Department of Civil Engineering and the Community Service Institution (LPPM Unsoed), which mobilized surveyors, students, and instruments to the disaster area. In addition, we highly appreciate Mr. Arvandi from the Agency of Public Works and Community Housing, Division of Municipal and Water Resources, Central Sulawesi Province for special assisstance during our activities in the Palu Bay area.

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

Table 1 Measured sites (see also Figures 1 and 2)

| No. | Site name | Measurement time | Coordinates | | Inundation distance (m) | Measuredr unup height (m) | Correction | Corrected runup height (m) | Watermark |
|---|---|---|---|---|---|---|---|---|---|
| | | | Longitude | Latitude | | | | | |
| 1. | Donggala City | 12-Oct-18 4:34:08PM | 119.741313 | -0.663054 | 35.60 | 2.80 | -0.40 | 2.40 | BL, SD |
| 2. | Loli Dondo | 16-Oct-18 3:27:28PM | 119.776100 | -0.731612 | 79.06 | 3.70 | -1.25 | 2.45 | BB, BV, DS |
| 3. | Loli Saluran | 16-Oct-18 2:16:54PM | 119.794095 | -0.783867 | 141.94 | 3.43 | -1.30 | 2.13 | BB, BV, DS |
| 4. | Watu Sampu | 16-Oct-18 12:48:04PM | 119.810032 | -0.822144 | 104.01 | 7.79 | -1.40 | 6.39 | BL, BV, DD, EW |
| 5. | Tipo | 17-Oct-18 10:25:41PM | 119.829355 | -0.864574 | 138.50 | 8.54 | -1.45 | 7.09 | DS, EW |
| 6. | Silae | 17-Oct-18 3:08:35PM | 119.834315 | -0.874580 | 130.04 | 5.50 | -1.15 | 4.35 | DD, DS, GD |
| 7. | Lere | 15-Oct-18 2:30:19PM | 119.843401 | -0.885372 | 290.56 | 2.78 | -1.20 | 1.58 | DD, DS, GD |
| 8. | Besusu Barat | 16-Oct-18 8:20:46AM | 119.860210 | -0.887457 | 270.10 | 2.71 | -1.05 | 1.66 | BB, DD, DS, GD |
| 9. | Talise | | | | 279.63 | | -0.95 | | |
| | Talise 1 | 15-Oct-18 | 119.873739 | -0.876266 | | 2.83 | | 1.88 | BB, DS, GD, EW |
| | Talise 2 | 8:12:18AM | 119.874616 | -0.873833 | | 2.98 | | 2.03 | DD, DS, GD |
| | Talise 3 | | 119.874389 | -0.874440 | | 3.06 | | 2.11 | DD, DS, GD |
| | Talise 4 | | 119.874294 | -0.875004 | | 2.75 | | 1.80 | DD, DS, GD |
| 10. | Tondo | | | | 260.34 | | -1.70 | | |
| | Tondo 1 | 14-Oct-18 | 119.881499 | -0.844691 | | 12.20 | | 10.5 | DC, DD,DS, EW |
| | Tondo 2 | 12:58:26 | 119.880688 | -0.843981 | | 9.44 | | 7.74 | DC, DD, DS, EW |
| | Tondo 3 | | 119.881253 | -0.845850 | | 11.61 | | 9.91 | DC, DD, DS, EW |
| | Tondo 4 | | 119.880854 | -0.846571 | | 9.97 | | 8.27 | DC, DD, DS, EW |
| 11. | Layana | 14-Oct-18 | | | 511.68 | | -0.70 | | |
| | Layana 1 | 7:45:14 | 119.887135 | -0.822159 | | 6.83 | | 6.13 | DD, DS, GD, WW |
| | Layana 2 | | 119.883472 | -0.823863 | | 3.04 | | 2.34 | DD, DS, GD |
| 12. | Mamboro | 13-Oct-18 | | | 295.30 | | -1.15 | | |
| | Mamboro 1 | 13:43:47 | 119.879074 | -0.801753 | | 5.10 | | 4.15 | DC, DD, DS, EW |
| | Mamboro 2 | | 119.878349 | -0.800542 | | 5.30 | | 3.95 | DC, DD, DS, EW |
| 13. | Taipa: | | | | 103.50 | | -1.30 | | |
| | Taipa 1 | 17-Oct-18 | 119.858686 | -0.778698 | | 3.93 | | 2.63 | BV, DS, EW |
| | Taipa 2 | 8:56:31 | 119.859367 | -0.779472 | | 5.30 | | 4.00 | BV, DS, EW |
| | Taipa 3 | 17-Oct-18 9:01:46 17-Oct-18 9:55:28 | 119.859542 | -0.779995 | | 5.66 | | 4.36 | BV, DS, EW |
| 14. | Pantoloan | 17-Oct-18 | 119.857660 | -0.710840 | 150.55 | 3.44 | -0.95 | 2.49 | DS, EW, GD |





| No. | Site name | Date/Time | Lon | Lat | | | | | Evidence |
|---|---|---|---|---|---|---|---|---|---|
| | | 13:33:12 | | | | | | | |
| 15. | Wani | 17-Oct-18 12:34:23 | 119.841543 | -0.693099 | 206.58 | 4.61 | -1.10 | 3.51 | BL, DD, EW, MO |
| 16. | Lero | 17-Oct-18 11:27:51 | 119.812422 | -0.629011 | 90.50 | 2.75 | -1.45 | 1.30 | DS, EW, WW |
| 17. | Tanjung Padang | 18-Oct-18 12:43:01 | 119.803220 | -0.231612 | 107.00 | 2.20 | -1.20 | 1.00 | DS, EW, GD |
| 18. | Lende | 18-Oct-18 14:11:29 | 119.817232 | -0.185461 | 40.20 | 2.15 | -0.90 | 1.25 | EW, GD |

BB: broken tree branch; BL: boat on land surface; BV: brown vegetation; DC: debris caught; DD: debris deposition; DS: damaged structures; EW: eye witnesses; GD: garbage deposition; MO: marine-origin objects; SD: sediment deposition; WW: watermark on wall.

5 Table 2 Land use and damage for each site

| No. | Site name | Land use | Damage |
|---|---|---|---|
| 1 | Donggala City | Fishing port, passenger and cargo port, urban area | Damaged houses, fisherman boat lifted to land |
| 2 | Loli Dondo | Settlement, fishery | Damaged houses |
| 3 | Loli Saluran | Settlement, stone mining | Damaged houses |
| 4 | Watu Sampu | Indonesian Navy harbour, agriculture | Navy vessel lifted to land |
| 5 | Tipo | Settlement | Damaged houses |
| 6 | Silae | Urban area, settlement | Damaged houses, |
| 7 | Lere | Urban area, business | Damaged mall, campus, offices |
| 8 | Besusu Barat | Urban area, offices, business | Collapsed 300-msteel bridge |
| 9 | Talise | Urban area, sightseeing, aquaculture | Damaged coastal garden, restaurants |
| 10 | Tondo | Settlement, sight seeing | Damaged houses |
| 11 | Layana | Warehouse, stores complex | Damaged warehouses and stores |
| 12 | Mamboro | Settlement | Damaged houses |
| 13 | Taipa | Passenger port, sight seeing | Damaged passenger terminal |
| 14 | Pantoloan | Passenger and cargo port | Washed away container |
| 15 | Wani | Fishery port, aquaculture | Ship lifted to land, severely damaged houses |
| 16 | Lero | Settlement, agriculture | Houses sunk by liquifaction |
| 17 | Tanjung Padang | Agriculture | Damaged houses and crops |
| 18 | Lende | Agriculture | Damaged houses and crops |



Table 3 Surface-earth landslide location

| No. | Location (village) | Longitude | Latitude | No. | Location (village) | Longitude | Latitude |
|---|---|---|---|---|---|---|---|
| 1 | Oti | -0.38700 | 119.76059 | 9 | Sindue Tombusabora | -0.51684 | 119.76921 |
| 2 | Oti | -0.39071 | 119.76287 | 10 | Sindue Tombusabora | -0.51763 | 119.76820 |
| 3 | Batusuya | -0.46021 | 119.76296 | 11 | Sindue Tombusabora | -0.51866 | 119.76789 |
| 4 | Batusuya | -0.46681 | 119.76296 | 12 | Sindue Tombusabora | -0.51909 | 119.76753 |
| 5 | Sindue Tombusabora | -0.51077 | 119.76903 | 13 | Sindue Tombusabora | -0.52031 | 119.76760 |
| 6 | Sindue Tombusabora | -0.50989 | 119.77216 | 14 | Sindue Tombusabora | -0.52549 | 119.77046 |
| 7 | Sindue Tombusabora | -0.51189 | 119.77101 | 15 | Sindue Tombusabora | -0.53106 | 119.77601 |
| 8 | Sindue Tombusabora | -0.51558 | 119.76896 | 16 | Sindue Tombusabora | -0.53503 | 119.77810 |





**Figure 1:** Survey area of Palu Bay located on Sulawesi island. A camera in a moving car was used to record the tsunami-affected area following the Trans Sulawesi Road paralleled Palu Bay coastline from Site 1 to Site 18.



**Figure 2:** Evidence of tsunami runup and inundation. (a) Debris left behind in the residential area of Tondo, (b) debris caught in a tree in Mamboro, (c) and (d) debris stuck in a tree in Tondo, (e) leaves turned brown due to being submerged in salt water, (f) a tree had green leaves at the top and brown at the lower part, indicating the tsunami inundation height limit in Layana, (g) debris lodged on top of a





building, (h) broken house element showing tsunami water level, (i) watermark on a house wall in Lero village, (j) sand deposit on building floor in Donggala City, (k) a 45-m-long ship moved to land in Wani harbour, (l) interview with an eyewitness in Mamboro.

**Figure 3:** Measurement results of inundation distances.





**Figure 4:** Measurement results of runup heights.

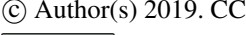



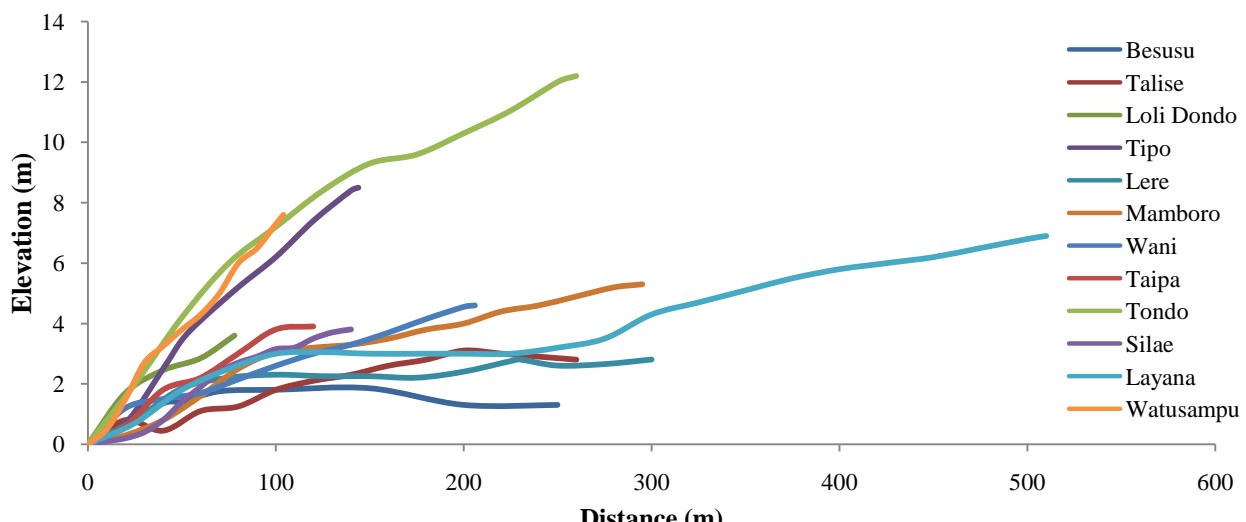

**Figure 5:** Transects of beach where tsunami wave arrived. The longest inundation distance is at the Layana site and the highest runup is at the Tondo site.

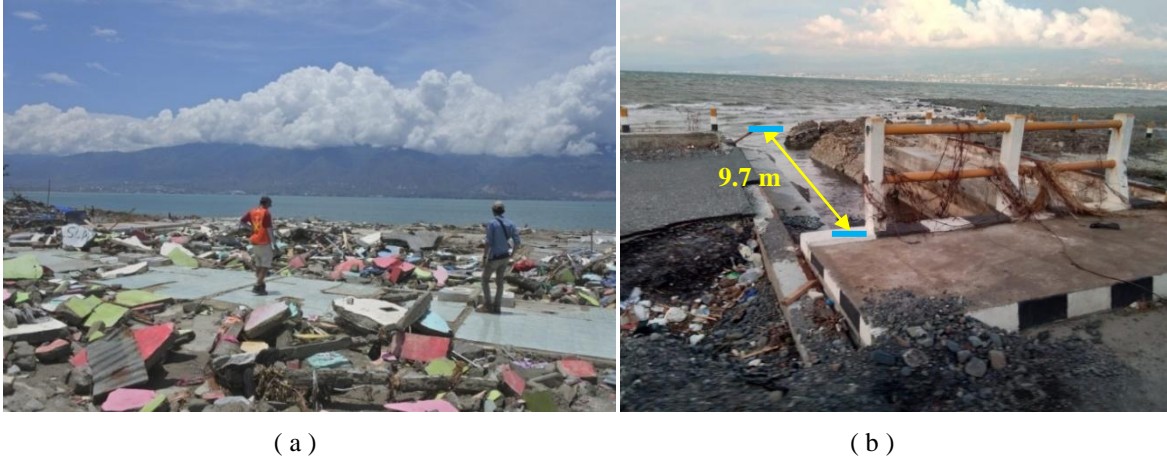

( a )                                      ( b )

**Figure 6: (a)** Damage caused by the tsunami in Tondo, a residential complex where a lot of private boarding houses were inhabited by students at the University of Tadulako, and **(b)** a reinforced concrete bridge on Cumi-cumi Road Palu City shifted by 9.7 m by the tsunami.



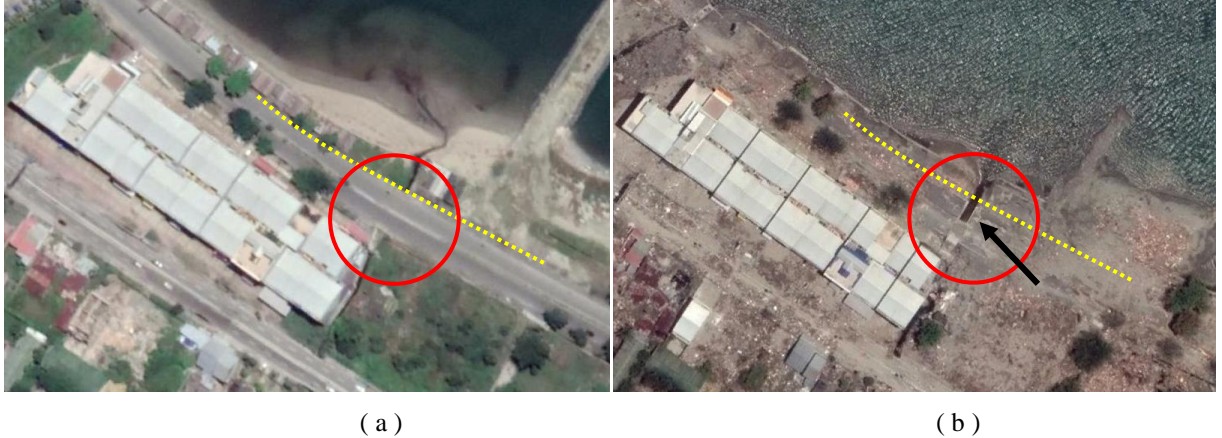

( a )                                                    ( b )

**Figure 7:** Satellite images taken on (a) September 26, 2017 and (b) October 2, 2018, showing the bridge shift.

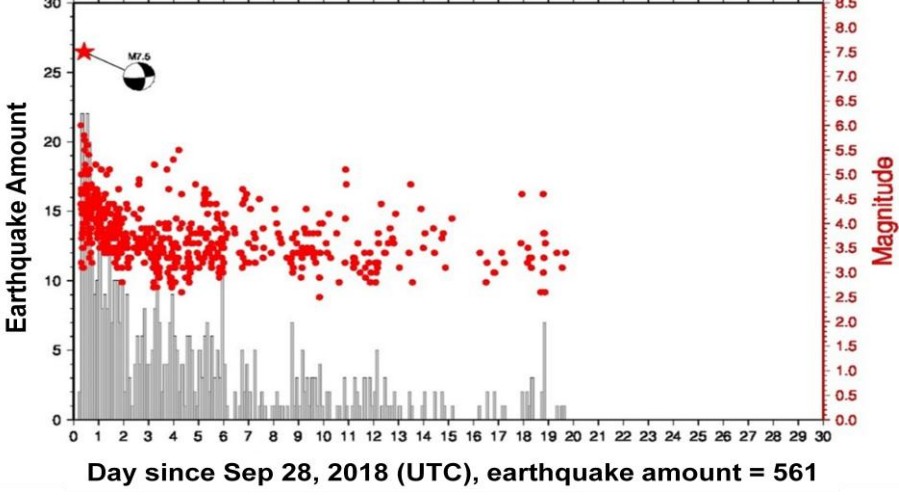

5    **Figure 8:** Aftershock earthquakes ($n = 561$) 20 days after the main earthquake.



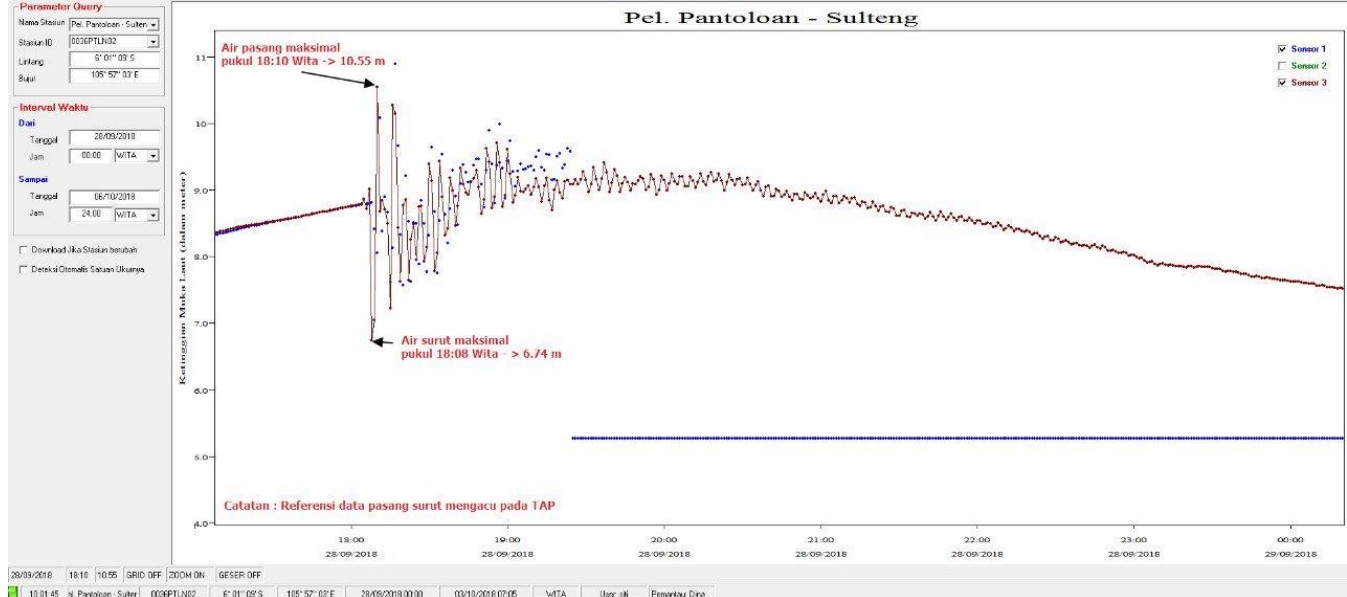

**Figure 9:** Water level recording at the Pantoloan tidal station (Courtesy: Sudibyo, 2018).

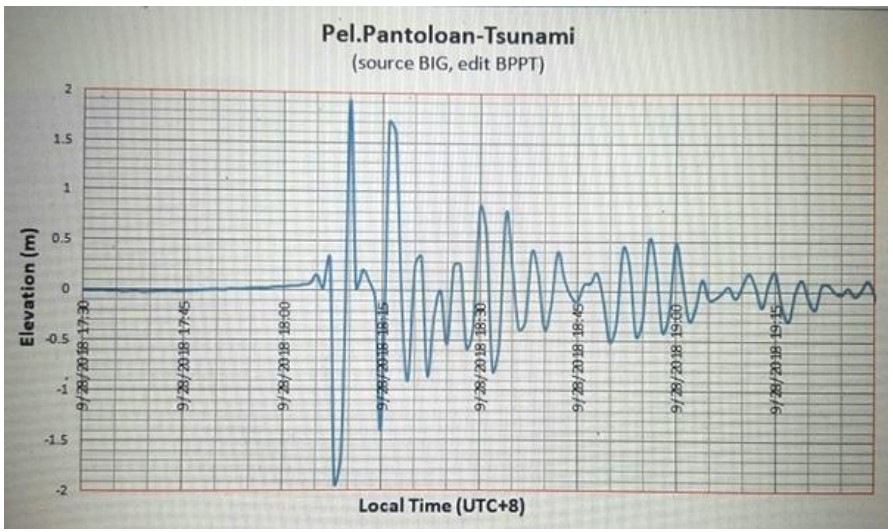

**Figure 10:** Magnified view of Fig. 9 (Courtesy: Indonesian Geospatial Agency and Agency for Assessment, Implementation of Technology).



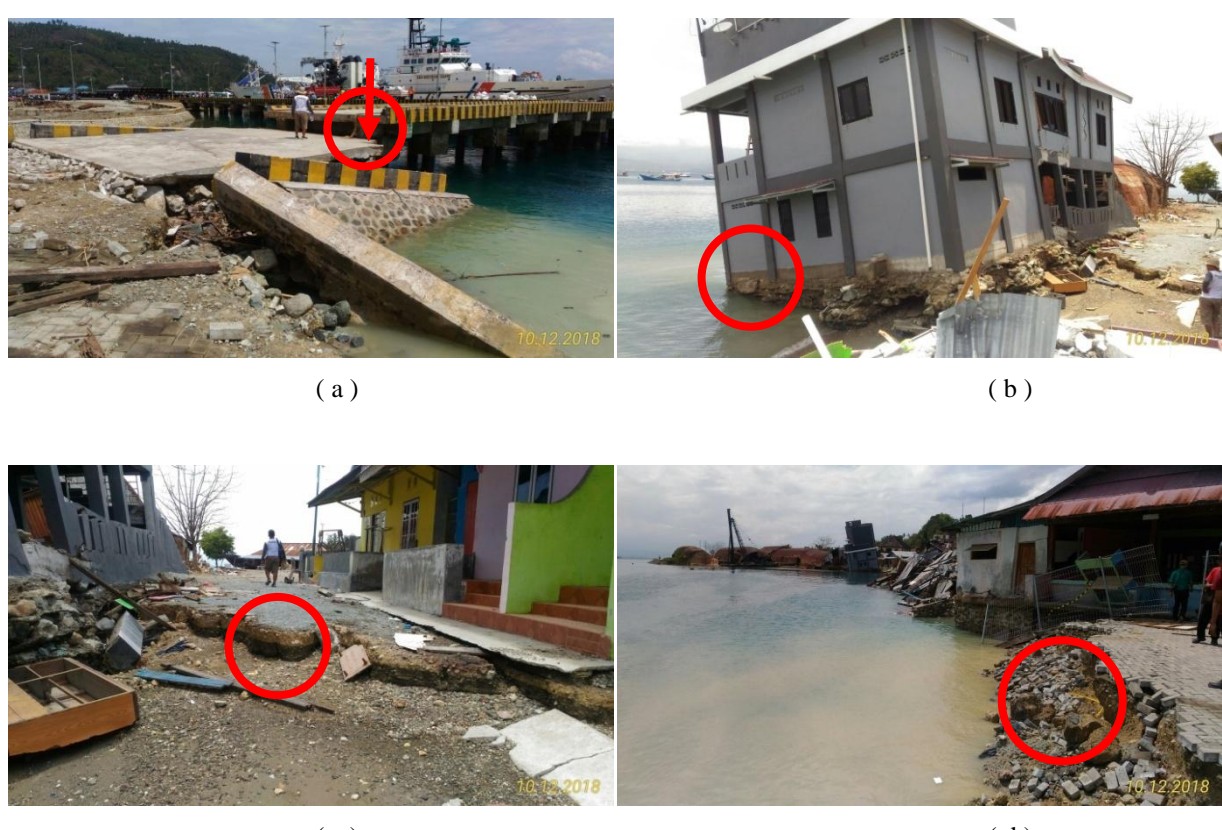

**Figure 11:** Landslide in Donggala City. (a) A trestle dropped 0.8 m in Donggala Port, (b) a building on the seaside slip down significantly, (c) the surface of an alley in a settlement dropped 0.4 m, and (d) a layered courtyard with paving blocks dropped around 1.5 m.

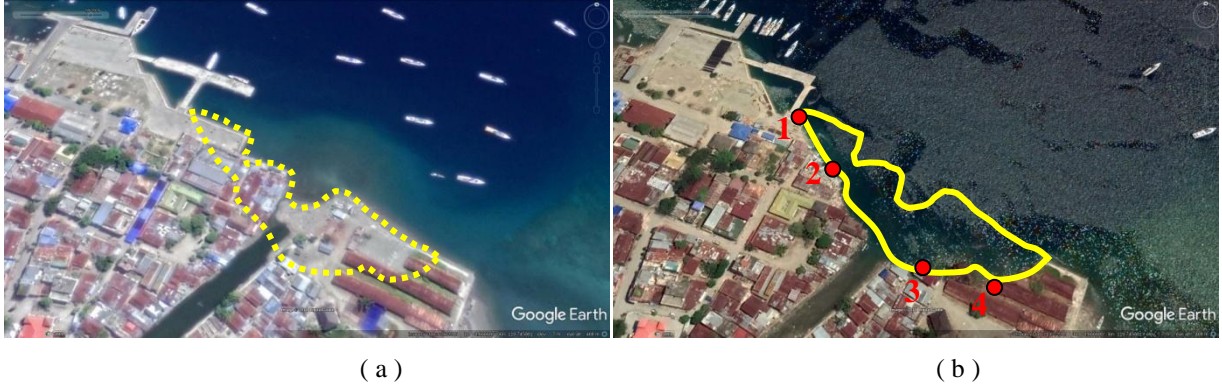

**Figure 12:** Possible landslide areas in Donggala (yellow dotted lines). Images were obtained from Google Earth. Satellite images taken on(a) 6 July, 2016 (more than a year before the earthquake) and (b) 2 October, 2018 (4 days after the earthquake and tsunami). The yellow bounded area is around 10,068 m$^2$ or 1 hectare. Number 1, 2, 3 and 4 in Fig. 12 b corresponds to Fig. 11 (a), (b), (c), and (d).



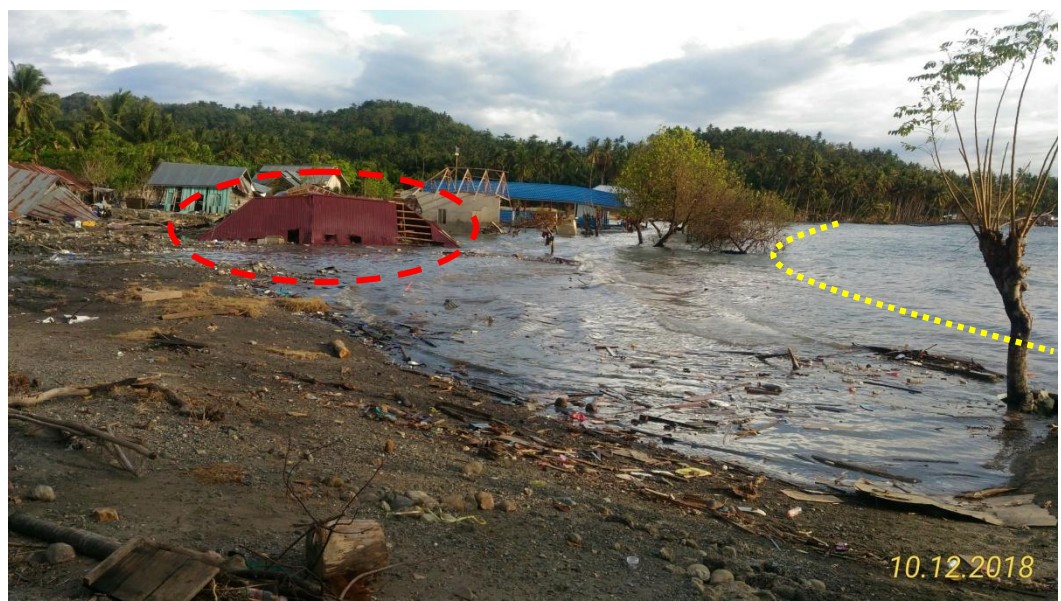

**Figure 13:** Quick landsubsidence in Lero Village. Photograph taken two weeks after the event. Some houses dropped suddenly, around 3-4 m, when the earthquake occurred. Residents of these houses, especially that indicated by the oval, could not save themselves. The yellow dotted line is the former coastline.

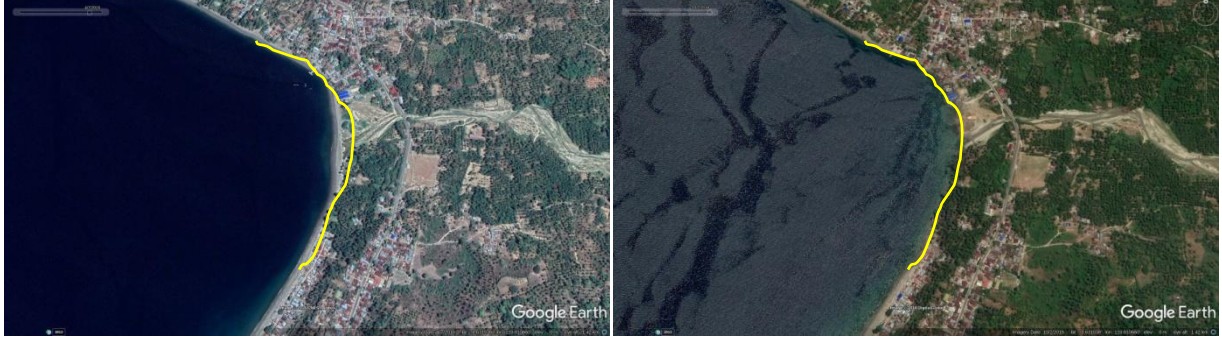

**Figure 14:** Quick land subsidence in Lero Village. Satellite images taken on (a) 7 April, 2016, and (b) 2 October, 2018, from Google Earth, showing conditions after the earthquake and tsunami. The area of land that dropped is 22,971 $m^2$ or almost 2.3 hectares.