# Peer review of "Post-event Field Survey of 28 September 2018 Sulawesi Earthquake and Tsunami"

_Natural Hazards and Earth System Sciences, 2019_

## Referee Comment (RC1) · Anonymous Referee #1 · 28 May 2019

Overall This is an interesting paper that describes the results of field work after the 2018 Sulawesi tsunami. The paper follows the general pattern of field work papers, and is important that such events are properly documents, so that modellers can then attempt to reproduce them. However, the paper suffers from lower than expected writing quality. The English is ok in some places, and poor in others. Also, the authors often repeat themselves. The more serious problem, however, comes from its unclear focus. Most of the paper deals with the tsunami damage, but at times the authors randomly include other information relating to aftershocks or landslides that are not related to the tsunami. Thus, several parts of the paper should be deleted, and the message should become more focused. Instead, the authors might want to describe the mechanisms of tsunami damage in more detail at each location (currently they only

superficially describe some locations).

Major comments

-By now a number of other field work papers have been published. Please find these, and cite them. Also, please explain what differences there are between your work and other papers P3, L24. In what way did the authors do this? How can they choose one point that can be representative for a tsunami that was as complicated as the one in this case? The English in the paper needs to be improved. In places the sentences are correct, and in others they are pretty poor. P4 L4-5 what kind of camera was used? Did the authors obtain a 360 degree view? Otherwise, in what way is this similar? Is this going to be opened to other researchers? (if not, what is the point of writing this?) P4 L19 and onwards. What is the point of talking so much about the rain, if the authors then dismiss the importance of it? P5 L2. Where all these measurements corrected for tide? Using which software? Are the datasets given in this paper those corrected for tide, or the original measurements? Also P5 L7-9, the location of these tidal stations needs to be shown in some figure. See also P5 L22, which indicates both corrected and uncorrected, making it unclear what the other numbers in the paper actually are. P6 L16. What is the point of this section? You are talking about earthquake damage, but this paper up to now is mostly about tsunami damage. Hence, it feels rather odd. I suggest just focusing on the damage by the tsunami, and delete this section. P6 L24 If the bridge was shifted, it was damaged. Not sure what the authors are trying to say here... Also, how can the authors say the area is only 3.4m2, given the description earlier? This part feels rather confusing. P7 L2. From which sites? What is the point the authors are trying to make here? P7 L4-10. What is the point of this talk of aftershocks? I suggest all this is deleted, and the authors focus just on the tsunami damage. Same for P8 L10-19 P8 L20-30 What is the point the authors are making here? The authors don't seem to conclude anything, and merely state conjecture. This might be ok if it was in the discussion section of the paper, but this is not it. P9 L1-5 What is the point in a scientific paper of stating that surveys are being carried out?

The authors should provide details or analysis, or let others do so. Reporting that something is happening is journalistic. P9 L19-22. There are already papers that are describing the location of landslides. Also, it is strange that the authors conclude this when they did not talk about this at length in their own paper (they should focus on the conclusions that can be derived from their own work).

Minor comments P2 L26 "Most of the victims came from"... P2 L30 astonished should not be used in academic literature P2 L30, by now you said many times that the earthquake took place due to an active strike-slip fault in Indonesia. Please delete P2 L32, again, you repeated many times that the earthquake destroyed many buildings P3 L11 "for a numerical model" P3 L12 "rebuilding of the affected areas by the 2018..." P3 L30 what is the point of saying that the authors took videos. Are these provided in the present research or any additional information? Otherwise delete... P4 L1 you repeated already that this road runs parallel to the coastline. P4 L9-10, delete these lines. P4 L17 "until the date of the end of the survey..." P4 L28 "The authors obtained important information from the surveys"... P4 L31 "The first wave acted as a trigger for evacuation, with many people starting to escape from the coastline". The technical word is "trigger". Please read other papers about evacuation triggers for tsunamis P5 L5 "is recorded at the maximum horizontal inundation distance". (Delete "the horizontal distance flooded by the wave) P5 L10 what do the authors mean by tsunami border? P5 L 12 Delete sentence starting by "The items scattered" P5 L17 what is "tsunami creeping"? run-up? P5 L23 "This area was flattened by the tsunami (Fig 6a), with no buildings surviving"? L5 L26 Rephrase "being quite nimble" P5L27 what exactly is this important observation? Be specific. P5 L32 tsunami risk managers know they can use this data for run-up modelling. Please delete this sentence, it is obvious. Same for P6 L31-32. P5 L18 check reference, not shown P8 L23 the words "impacted areas with a relatively narrow width" are unclear. Revise. P8 L28 "Ulrih et al. (2019) assume that a..."

[Figure]

2019-91, 2019.

---

## Short Comment (SC1) · 16 Jun 2019

Tsunamis are rare and have different characteristics each other. So that a field survey is essential. A short comment, I think authors might plot the epicenter point in their figures, although its coordinate has been mentioned in the text. Then reader will know another uniqueness of Palu tsunami that the epicenter was in land, not undersea. Beside, author might improve the map of survey sites point, runup and inundation distance become neater (Fig 1, Fig 3 and Fig 4).

---

## Referee Comment (RC2) · Ahmet Cevdet Yalciner (Referee) · 25 Aug 2019

Page-1 Line 14: Indicating the name of the university mentioned would provide more clear information. Line 22-23: Do the authors have any reference for the earthquake parameters given? Line 23-24: Do the authors have any reference for the numbers reported?

Page-3 Line 2-5: "For tsunamis, post-incident surveys are often carried out. Major tsunamis such as. . .." Only stating some of the tsunami post-event surveys like "giving examples" may not be appropriate. Two suggestions: Either state the importance and relation of them with this study OR delete these sentences. Line 9: "Observation of damage was also conducted." Too general sentence. What kind of damage data is

collected? Any details on the data collection processes?

Page 9: Line 14: "There were three main tsunami waves that reached the beach." Which beach? Not clear. Line 14: "The first wave was relatively low." With respect to what? You should State it more clearly and in an understandable way. Line 18: "The wave that hit the beach was quite high." This sentence by itself does not provide any meaningful information.

General Comments: - A brief summary and citation of previously published papers on 2018 Palu Event field survey is necessary. - Section 5.1, Aftershock information is not related with the focus of this study and the work done. - The conclusion section should be rewritten by clear sentences and providing a comprehensive summary of the results obtained. For example, Giving ranges such as "2 to10 m and the inundation distances were 80 to 500 m." Or "The arrival time of waves varied from 3 to 10 minutes.." does not provide satisfying information. The authors, at least, may add the locations of these measurements.

---

## Author Comment (AC1) · 19 Sep 2019

Referee 1 – Authors Interaction We would like to thank Anonymous Referee 1 for the constructive comments and suggestions towards improving our manuscript. Our response was also uploaded in the form of a supplement. We summarize comments from Referee 1, author's response, and author's changes in manuscript as follows.

Comment 1: Overall This is an interesting paper that describes the results of field work after the2018 Sulawesi tsunami. The paper follows the general pattern of field work papers,and is important that such events are properly documents, so that modellers can then attempt to reproduce them. However, the paper suffers from lower than expected writing quality. The English is ok in some places, and poor in others. Also, the

[Figure]

authors often repeat themselves. The more serious problem, however, comes from its unclear focus. Most of the paper deals with the tsunami damage, but at times the authors randomly include other information relating to aftershocks or landslides that are not related to the tsunami. Thus, several parts of the paper should be deleted, and the message should become more focused. Instead, the authors might want to describe the mechanisms of tsunami damage in more detail at each location (currently they only superficially describe some locations).

Response 1: Thanks for very detailed and constructive comments from Referee1. We are improving our writing quality, including English. An native at an English proofreading service center would handle our manuscript. That's right we often repeated ourselves,now we reduce it. We modify the writing in order to more focus by removing several parts you suggest, e.g aftershock and earth-surface landslide. Nevertheless, we preserve part about coastal landslides since we feel to have contribution on it. We strongly agree with your advice on description of the mechanism of tsunami damage. So that we add description on runup, inundation, and damage at each site in our text. It could be seen on uploaded supplement file for final response.

MAJOR COMMENTS

Comment 2: By now a number of other field work papers have been published. Please find these,and cite them. Also, please explain what differences there are between your work and other papers.

Response 2: Alright. We are adding other field work papers, i.e Omira et al (2019),Mikami et al. (2019), Muhari et al. (2018),Yalciner et al. (2018), Putra et al. (2019), Sassa & Takagawa (2019), Takagi (2019), Arikawa et al. (2018). The last three papers focus on coastal landslides, while Putra et al. evaluate runup based on tsunami deposit. Muhari et al. was probably the first team coming in disaster area. They gave early report as direction for other team coming later. Their results are preliminary and limited around Palu City, in the end of Palu Bay. The most close topic with us is by

[Figure]

Omira et al. (UNESCO international team) and Mikami et al.. They measured more points than ours. Our several points intersect with their points. Nevertheless, they did not measure inundation distance as done by our team. We cite them and explain it in the manuscript.

Change in manuscript 2: P3 L5-12 supplement Many groups have carried out field surveys of the Sulawesi tsunami event or also known as Palu tsunami. Muhari et al. (2018) investigated wave height and inundation depth at several points with a focus around the end of the bay. A UNESCO international tsunami survey team surveyed 125 km of coastline along the Palu Bay up to the earthquake epicentre region. The team performed 78 tsunami runup and inundation height measurements throughout the surveyed coastline (Omira et al., 2019; Yalciner et al., 2018). Putra et al (2019) focus more on tsunami deposits. Meanwhile, Arikawa et al. (2018), Sassa and Takagawa (2018), and Takagi et al. (2019) each conducted a survey related to coastal subsidence, coastal liquefaction or submarine landslide detected in Palu Bay. This survey data can be combined with data from other groups, especially we contribute to provide data of runup height, inundation distance, flow depth, and damage at different points and coordinates.

Comment 3: P3, L24. In what way did the authors do this? How can they choose one point that can be representative for a tsunami that was as complicated as the one in this case?

Response 3: P3 L24 We modify the sentence. We mean that we chose some sites (not only one point) which had significant impact caused by the tsunami. We measured runup, inundation, and flow depth at 18 sites. On the first day of our survey we recorded situations along the Palu Bay. From these recordings we can roughly estimate sites with high runup and the long inundation. These points usually also have a severe level of damage. Besides, important places such as ports and densely populated areas are our priority. Measuring a coastal cross section at each certain distance, for example, every 1 km along 70 km of Palu Bay, might provide more representative data, but we

have difficulty doing that mainly because it will take a long time. In addition, the areas affected by the tsunamis were also fragmented, not connected.

Change in manuscript 3: P3 L25-26 supplement ..... 2) tracing the road along the coast in Palu Bay to get an overview of the affected area; 3) choosing some sites that had significant impact of the tsunami; 4) looking for evidence of runup boundaries .....

Comment 4: The English in the paper needs to be improved. In places the sentences are correct, and in others they are pretty poor.

Response 4: Thanks for the assessment. We are trying to meet referee's suggestion, improving English in the manuscript thoroughly, and it would be checked by English proofreading service center.

Comment 5: P4 L4-5 what kind of camera was used?Did the authors obtain a 360 degree view? Otherwise, in what way is this similar? Is this going to be opened to other researchers? (if not, what is the point of writing this?)

Response 5: P4_L4-5 We delete "This method is similar to that used by Google Street View$^{®}$, but we used simpler equipment." The idea may be same with Google Street View$^{®}$ but the method and camera used was not same. However, we think our video collections are useful. We plan to put them in the supplement in order to be opened to other researchers. We use Google Street View$^{®}$ for comparing with our videos to evaluate damage along Trans Sulawesi Road.

Changes in manuscript 5: P4 L6-7 supplement ..... A camera on a moving car was operated to record the situation around the coastal area. It produced a number of videos describing the damage (contained in supplement). P5 L6-7 response supplement .....Video recorded along trans Sulawesi Road were compared to Google Street View, Google Map, and Google earth in order to assess the distance of damage from coastline. ....

Comment 6: P4 L19 and on wards. What is the point of talking so much about the rain,

if the authors then dismiss the importance of it?

Response 6: P4_L19 We replace "Fortunately, from the point of view of conducting a survey, surface runoff due to rain seems insignificant and does not erase the tsunami footprint." with "It was a challenging work to look for tsunami footprint on surfaces that were exposed to surface runoff." In addition, we also shorten the paragraph containing about rain.

Change in manuscript 6: P4 L14-19 supplement ..... October is the beginning of the rainy season in Indonesia, including Sulawesi. Palu City is located near the equator, as shown by latitudes in Table 1. It is one of the driest areas in Indonesia, with rainfall recorded at the Mutiara Meteorology Station in 2017 of 774.3 mm. Since the earthquake incident until the date of the end of the survey, it rained four times, three of which occurred during our survey period, with a duration of less than 2 hours and with low to moderate intensity. It was a challenging work to look for tsunami footprint on surfaces that were exposed to surface runoff caused by rains. .....

Comment 7: P5 L2. Where all these measurements corrected for tide? Using which software? Are the data sets given in this paper those corrected for tide, or the original measurements? Also P5 L7-9, the location of these tidal stations needs to be shown in some figure. See also P5 L22, which indicates both corrected and uncorrected, making it unclear what the other numbers in the paper actually are.

Response 7: P5 L2 Runup heights were corrected to calculate heights above sea level at the time of survey by using WXTide software version 4.7, available at www.wxtide32.com/index.html. We used Donggala station listed in the software for correcting and assume no significant variations on the sea level inside Palu Bay.We modified that all number (runup and inundation) shown in the paper are corrected for tide.Thanks for rigorous comments. P5_L7-9 Authors ploted Donggala (replacing Mamuju) and Pantoloan stations in some figures. P5_L22 We modified that all number (runup and inundation) shown in the paper are corrected for tide.

Changes in manuscript 7: P5 L1-3 supplement ..... Runup heights were corrected to calculate heights above sea level at the time of survey by using WXTide software version 4.7, available at www.wxtide32.com/index.html. We use Donggala station, the closest station listed in the software, for correcting and assume no significant variations on the sea level inside Palu Bay.

P5 L11-13 supplement ..... The measurement results are shown in Table 1 and Figs. 3-5. The measurement values in the table has been corrected with tides. Runup height and inundation distance vary from site to site.

P5 L28-29 supplement ..... A total of 4 cross sections of these coast were measured by our team. The measured runup heights were 10.73, 7.97, 10.14, and 8.50 m, respectively, as shown in Table 1. .....

Comment 8: P6 L16. What is the point of this section? You are talking about earthquake damage,but this paper up to now is mostly about tsunami damage. Hence, it feels rather odd. I suggest just focusing on the damage by the tsunami, and delete this section. P6 L24 If the bridge was shifted, it was damaged. Not sure what the authors are trying to say here: : : Also, how can the authors say the area is only 3.4m2, given the description earlier? This part feels rather confusing.

Response 8: P6_L2-6 and L12-16 We delete these parts and try to focus on the damage by tsunami as you suggesting. P6_L24 We revise its area = 244.7 m2, thanks for the precise comment. We measured size of bridge which moved from original position with intention to give data about bridge dimension that may be used by modeler to assess tsunami force (drag force, lift force, etc.). We could provide sketch of the bridge and put it in on supplement if it is needed.

Change in manuscript 8: P9 L2-3 supplement ..... Based on these dimensions, the surface area of the bridge was 244.7 m2, the volume was 23.4 m3, and the mass was approximated to be around 56 tons. .....

Comment 9: P7 L2. From which sites? What is the point the authors are trying to make here? P7 L4-10. What is the point of this talk of aftershocks? I suggest all this is deleted, and the authors focus just on the tsunami damage.

Response 9: P7_L2 We mean it from our measuring sites, Palu bay area. Additional data were documented beside runup and inundation measurement. But, we deleted it. P7_L4-10 We deleted section about aftershock in order to be more focused. Thanks.

Change in manuscript 9: Please see the supplement Comment 10: Same for P8 L10-19 P8 L20-30 What is the point the authors are making here? The authors don't seem to conclude anything, and merely state conjecture. This might be ok if it was in the discussion section of the paper, but this is not it.

Response 10: P8_L10-19 and P8 L20-30 has been removed to make more focused.

Change in manuscript 10: Please see the supplement.

Comment 11: P9 L1-5 What is the point in a scientific paper of stating that surveys are being carried out? The authors should provide details or analysis, or let others do so. Reporting that something is happening is journalistic. P9 L19-22. There are already papers that are describing the location of landslides. Also, it is strange that the authors conclude this when they did not talk about this at length in their own paper (they should focus on the conclusions that can be derived from their own work).

Response 11: P9_L1-5 Thanks for the advice. We deleted "Therefore, the Indonesian Navy deployed the KRI Spica Ship, ..... ..... ..... to conduct a bathymetry survey of Palu Bay after the tsunami." . P9_L19-22 We modified this part "Land subsidence in Donggala City (10,068 m2) and Lero Village (22,971 m2) gives evidence for underwater landslides. However, it does not mean there were only two underwater landslides; more landslide locations may be found in the disaster area. " to be "Coastal landslides detected by our team in Donggala City (lost surface area of 10,068 m2) and Lero Village (lost surface area of 22,971 m2) gives additional evidence towards coastal landslides

found by other team as reported by Arikawa et al. (2018) and Omira et al. (2019)."

Change in manuscript 11: P10 L2-5 supplement ..... Coastal landslides detected by our team in Donggala City (lost surface area of 10,068 m2) and Lero Village (lost surface area of 22,971 m2) gives additional evidences towards coastal landslides found by other team as reported by Arikawa et al. (2018) and Omira et al. (2019). .....

MINOR COMMENTS

Comment 12: P2 L26 "Most of the victims came from": : : P2 L30 astonished should not be used in academic literature. P2 L30, by now you said many times that the earthquake took place due to an active strike-slip fault in Indonesia. Please delete P2 L32, again, you repeated many times that the earthquake destroyed many buildings.

Response 12: P2_L26 We revised it to be"Most of the victims came from..." P2_L30 We replaced "astonished" with"surprised" P2_L30 We removed "The Palu-Koro fault which divides Sulawesi into two parts, has quite active tectonic activity...." ; Moved "the movement of ..." ; and "This is the second most active fault in Indonesia after the Yapen fault in Papua." ..... P2_L32 We deleted it and reduce repeated words or sentences.

Changes in manuscript 12: P2 L19 supplement ..... Most of the victims came from this city. .... ..... P2 L23 supplement This disaster in Central Sulawesi has surprised scientific community. For a strike-slip fault, the plates move horizontally and thus do not usually cause enough vertical deformation to trigger a huge tsunami. It is still in discuss whether the tsunami caused by co-seismic deformation or non-tectonic sources. Field surveys play an important role to support seeking answer for question arised. ..... P2 L5 supplement The movement of rock formations is 35-44 mm/year (Bellier et al., 2001)

Comment 13: P3 L11 "for a numerical model" P3 L12 "rebuilding of the affected areas by the 2018: : :" P3 L30 what is the point of saying that the authors took videos. Are these provided in the present research or any additional information? Otherwise delete: : :

Response 13: P3_L11 "for a numerical model"... P3_L12 We change ".…. rebuilding the affected areas of the 2018 Sulawesi Tsunami" with "rebuilding of the affected areas by the 2018..." P3_L30 We deleted "We also recorded videos and took photographs."

Changes in manuscript 13: P2 L34 - P3 L1 supplement ..... For instance, Lynett et.al (2003) employed the field survey data of the 1998 Papua New Guinea tsunami as validation for a numerical model, ..... P3 L3-44 supplement ..... rebuilding of the affected areas by the 2018... P3 L31-32 supplement ..... Therefore, our team searched for video recordings and photographs made by local residents while conducting the measurement survey.

Comment 14: P4 L1 you repeated already that this road runs parallel to the coastline. P4 L9-10, delete these lines. P4 L17 "until the date of the end of the survey: : :" P4 L28 "The authors obtained important information from the surveys": : : P4 L31 "The first wave acted as a trigger for evacuation, with many people starting to escape from the coastline". The technical word is "trigger". Please read other papers about evacuation triggers for tsunamis

Response 14: P4_L1 We reduced the repeated phrase "parallel to the coastline" and now only one. P4_L9-10 We deleted these lines "Many locations with steep cliffs and tsunami trails were not easily visible. We did not take measurements in such locations. Likewise, we did not measure places not significantly affected by the tsunamis." P4_L17 We modified "Since the earthquake incident until the date of our team's return " to be "Since the earthquake incident until the date of the end of the survey" P4_L28 We modified "We got some important information from the interviews to be "The authors obtained important information from the surveys" P4_L31 We modified "The first wave was a warning so many people went away from the coastline immediately" to be "The first wave acted as a trigger for evacuation, with many people starting to escape from the coastline".

Changes in manuscript 14: P4 L2-3 supplement ..... The road connecting the provinces

on Sulawesi island, called the Trans Sulawesi Road, is mostly parallel to the coastline of the bay. ..... P4 L16 supplement ..... Since the earthquake incident until the date of the end of the survey. ..... P4 L26 supplement ..... The authors obtained important information from the surveys, ... ..... P4 L29-30 supplement ..... The first wave acted as a trigger for evacuation with many people starting to escape from the coastline. .....

Comment 15: P5 L5 "is recorded at the maximum horizontal inundation distance". (Delete "the horizontal distance flooded by the wave) P5 L10 what do the authors mean by tsunami border? P5 L 12 Delete sentence starting by "The items scattered" P5 L17 what is "tsunami creeping"? run-up? P5 L23 "This area was flattened by the tsunami (Fig 6a), with no buildings surviving"? P5 L26 Rephrase "being quite nimble" P5L27 what exactly is this important observation? Be specific. P5 L32 tsunami risk managers know they can use this data for run-up modelling. Please delete this sentence, it is obvious.

Response 15: P5_L5 Done. We deleted"the horizontal distance flooded by the wave" P5_L10 It is meant limit of inundation. We deleted "from coastline to tsunami border on land", and made new sentences. P5_L12 Done. We delete the sentence and made new sentences in part 5 about damage observation. P5_L17 Right. We mean "creeping" was runup. We replaced "creeping" by "runup". P5_L23 We deleted "This area was flattened by tsunami (Fig. 6a), buildings collapse." And added "caused a few building surviving" in part 5 about damages. P5_L26 We replaced "being quite nimble" by "have agility to save ..." P5_L27 We deleted "This is an important observation for future mitigation efforts."

Changes in manuscript 15: P5 L10-11 supplement ..... In the simplest case, the runup value is recorded at maximum horizontal inundation distance (IOC Manuals and Guides No. 37, 2014). ..... P6 L1-2 supplement ..... Because of this sloping topography, the tsunami wave reached as far as 488 m inland. It was the longest distance among others. ..... P8 L15-16 supplement ..... This site is a trade complex that supports the economic activities of Palu City in particular and Central Sulawesi Province in general.

The buildings damaged at this site functioned as shops, warehouses, and corporate offices.

P5 L29-30 supplement ..... The runup height of 10.73 m is the highest in this survey (Fig. 5) caused a few building surviving. P8 L13-14 supplement ..... This is likely due to most of the young residents have agility to save themselves when the tsunami arrived.

Comment 16: Same for P6 L31-32. P6 L18 check reference, not shown P8 L23 the words "impacted areas with a relatively narrow width" are unclear. Revise. P8 L28 "Ulrih et al. (2019) assume that a: : :"

Response 16: P6 L31-32 We deleted "This phenomenon can be further analyzed to determine the tsunami force." P6 L18 OK, it refers to Fig 6b. P8_L23 We removed part 5.5 about underwater landslides so that we deleted paragraph contained "This suspected source should be located near the impacted areas with a relatively narrow width (Muhari et al., 2018).". P8_L28 We revised and move this part "Ulrich et al. (2019) assume that a ..." to introduction part.

Changes in manuscript 16: P9 L8-9 supplement ..... The position of this bridge is at the end of Palu Bay (-0.88123°S 119.83907°E). ..... P8 L32response supplement ..... A detail measurement was conducted to a reinforced concrete bridge on Cumi-cumi Road, Palu City (Fig 6b). It gives a clue regarding the tsunami's strength. .... P2 L25-26 supplement ..... Ulrich et al. (2019) assume that a source related to earthquake displacements is probable and that landsliding may not have been the primary source of the tsunami. .....

Please also note the supplement to this comment:
https://www.nat-hazards-earth-syst-sci-discuss.net/nhess-2019-91/nhess-2019-91-AC1-supplement.pdf

**Supplement:**

[revised manuscript text omitted]

model, i.e. Boussinesq model and a nonlinear shallow water wave model. Yalciner (2001) conducted field survey and modeling of the 1999 Izmit tsunami which the location had similarity in geographycal feature, earthquake magnitude and tsunami mechanism with recent Sulawesi case. More broadly, these data can be used for disaster mitigation and rebuilding of the affected areas by the 2018 Sulawesi Tsunami.

5   Many groups have carried out field surveys of the Sulawesi tsunami event or also known as Palu tsunami. Muhari et al. (2018) investigated wave height and inundation depth at several points with a focus around the end of the bay. A UNESCO international tsunami survey team surveyed 125 km of coastline along the Palu Bay up to the earthquake epicentre region. The team performed 78 tsunami runup and inundation height measurements throughout the surveyed coastline (Omira et al., 2019; Yalciner et al., 2018). Putra et al (2019) focus more on tsunami deposits. Meanwhile, Arikawa et al. (2018), Sassa and

10  Takagawa (2018), and Takagi et al. (2019) each conducted a survey related to coastal subsidence, coastal liquefaction or submarine landslide detected in Palu Bay. This survey data can be combined with data from other groups, especially we contribute to provide data of runup height, inundation distance, flow depth, and damage at different points and coordinates.

**2 Survey Details**

A team from National Cheng Kung University, Taiwan, and Universitas Jenderal Soedirman, Indonesia, arrived at Sis Aljufri

15  Airport in Palu City at 06:00 a.m. Central Indonesia Time on October 11, 2018, thirteen days after the tsunami event. Studies have shown that surveys can be carried out successfully within two to three weeks of an event (Synolakis and Okal, 2005). Starting from the afternoon of October 11, a field survey was conducted until October 19 evening, for a survey period of 9 days. The emergency response period for the disaster area was determined by the Indonesian government to be one month (September 28 to October 26, 2018). The victim evacuation period was two weeks (September 28 to October 12). This

20  means that the survey was conducted in the emergency response stage, one day before the victim evacuation period ended. During this period, cleaning of area impacted by tsunami was still in progress, so that debris could be seen in the disaster area.

[revised manuscript text omitted]

Runup heights were corrected to calculate heights above sea level at the time of survey by using WXTide software version 4.7, available at www.wxtide32.com/index.html. We use Donggala station, the closest station listed in the software, for correcting and assume no significant variations on the sea level inside Palu Bay.

Damage observation was carried out at each site of surveys. We focus on damage of building and infrastructures although other kind of damage are interesting, such as vegetation, shoreline, and properties (cars, boats, fisherman tools, etc.). Videos and photographs were produced to assess the damage. Video recorded along trans Sulawesi Road were compared to Google Street View, Google Map, and Google Earth in order to assess the distance of damage from coastine. In addition, detail measurement of dimension was done for special object (for instance bridge) which is useful for tsunami force analysis.

**3 Inundation and Runup Measurements Result**

Runup is the maximum ground elevation wetted by the tsunami on a sloping shoreline. In the simplest case, the runup value is recorded at maximum horizontal inundation distance (IOC Manuals and Guides No. 37, 2014). The measurement results are shown in Table 1 and Figs. 3-5. The measurement values in the table has been corrected with tides. Runup height and inundation distance vary from site to site.

Western coast of Palu Bay comprises of Site 1 to 6. Site 1 (Donggala City) is located at the mouth of the bay. Runup height and inundation distance at this site were not significant. Site 2 and 3 namely Loli Dondo Village and Loli Saluran Village where had runup height for both sites is relatively the same, 2.53 m and 2.18. Inundation distance were short due to steep hills towards the mainland. Site 4 and 5 (Watusampu Village and Tipo Village) had height of runup, 6.63 m and 7.79 m. The inundation distances were 71.51 m and 91.11 m. High runup with short inundation was influenced by steep topography. The highest runup for western coast was found in Tipo (7.79 m), followed by  Watusampu (6.63 m).

The site at the southern coast of the bay (end of Palu Bay) consists of sites 7 to 9 (Lere, Besusu Barat, and Talise). The runup heights at these site were low at 1.40 m and 1.12 m for Lere and Besusu Barat. Talise had a higher runup of 3.02 m, but all the three had almost the same inundation distance between 200 to 250 m. The density of buildings in this area seems to have prevented the tsunami from reaching further inland. Flat topography caused runup elevation that did not differ much from sea levels.

Survey sites in the eastern coast area of Palu Bay consists of Site 10 to 16. Site 10 was located in Tondo. The topography of this area is relatively steep with a slope of 0.06 (6%). Evidence of tsunami water rise was in the form of debris on top of buildings, truncated building elements, collapsed walls, trash carried away, and fixed debris. A survivor also showed the highest places of tsunami water in this area. A total of 4 cross sections of these coast were measured by our team. The measured runup heights were 10.73, 7.97, 10.14, and 8.50 m, respectively, as shown in Table 1. The runup height of 10.73 m is the highest in this survey (Fig. 5) caused a few building surviving. Omira et al. (2019) shows that the highest runup from their field survey was in Benteng Village with height of 9.1 m. Benteng Village (in western coast) is viz-a-viz with highest runup location of our survey in Tondo (in eastern coast).

North of Tondo is Site 11 (Layana). The topography of this site is relatively flat with a slope of 0.013 (1.3%). Because of this sloping topography, the tsunami wave reached as far as 488 m inland. It was the longest distance among others. The runup points reached 6.57 m and 2.78 m in this site. Both varied greatly because many buildings have long and wide walls that stemmed the tsunami flow further inland.

Site 12 and 13 are Mamboro and Taipa. Runup height of 3.5 m and flow depth of 5.36 m caused severe damage houses and casualties in Mamboro. In Taipa, runup of 4.88 m reached the roof of passenger terminal of Taipa port. North of Pantoloan port is Wani port (Site 15). Runup, inundation and flow depth were significant at this site, 3.58 m and 5.12 m respectively. Site 16 (Lero) is northernmost of survey site inside Palu Bay. This site is face-to-face with Site 1 where are also lies in mouth of Palu bay. The two last sites were Tanjung Padang and Lende. These site located outside of Palu Bay and close to the epicenter. Around 1 m runup hit the both site. Coastal area between site 16 and 17 is steep slope of hilly area and no tsunami footage existed.

**4 Tsunami Arrival Time**

Arrival time of a tsunami wave is one of the main parameters calculated in tsunami modeling. The time needed for the tsunami wave to propagate from earthquake source location to the coast is defined by the estimated time of arrival (ETA) (Strunz et al., 2011). It is important related to early warning system.

Tidal records may provide a clue on tsunami arrival time. The tidal station closest to the disaster site is Pantoloan Tidal Station. This station is located inside Palu Bay, on a pier in Pantoloan Port and operated by the Agency of Geospatial Information. When the earthquake and tsunami occurred, the recording equipment was not damaged but the data transfer stopped because the communication network was interrupted. Fig. 8 and Fig. 9 shows the water level recorded when the tsunami arrived. The maximum low tide (6.74 m) was at 18:08 local time and the maximum tide (10.55 m) was at 18:10 local time. This means that the tsunami wave height recorded at the station was 3.8 m. This wave height can be seen more clearly in Fig. 9, which is from the same source as that for Fig. 8. In addition, the first tsunami wave arrived at 18:07, with the wave trough at 18:08 and the wave crest at 18:10 local time (UTC+8).

Other hint regarding tsunami arrival time are based on videos on social media, internet, and television, as well as eyewitnesses. More than one tsunami wave hit the coastal zone in Palu Bay. Most witnesses stated that three tsunami waves had arrived. The first was less than 1-m high. The second and third waves were much higher, and were quantified by measurements in this survey. The number of tsunami waves and their height order were similar to the 17 July 2006 Tsunami in Java. That event also had three tsunami waves which the first one was of little magnitude and was followed by the second wave which was the highest one (Lavigne et al., 2007). Witnesses did not give an exact arrival time of the tsunami wave on the coastal zone in Palu Bay. Generally, they referred to prayer times as a guide. Indonesia is majority Muslim. The time of the earthquake and tsunami is close to one of the Muslim worship times in the afternoon, which coincides with a sunset

called "maghrib" prayer. The prayer schedule circulated by the Ministry of Religion of the Republic of Indonesia for the area of Palu City and Donggala Regency indicates that the starting time of "maghrib" prayer period on September 28, 2018, was 17:58 local time. Normally, there are two call sounded from a mosque as starting time sign for praying. The first call is called "adzan" and the second call is called "iqamah". The period between the two call is 10 minutes. Some news, videos,

5   and witnesses show that the tsunami came when people were preparing to pray, between "adzan" and "iqamah". The $M_W = 7.5$ earthquake occurred at 18:04. This shows that the tsunami waves came less than 10 minutes after the earthquake or between 18:05 and 18:15 local time, different for each site in the disaster area. Thus, the testimony of the witnesses was consistent with the detection of tidal gauges at the Pantoloan station. The important note from the September 2018 event is that the tsunami arrival time was very short.

10   **5 Buildings and Infrastructures Damage**

We identified damages to buildings and structures caused by the Sulawesi event 2018 into 3 types, namely damage due to earthquakes, liquefaction, and tsunamis. Damage caused by earthquakes is characterized by horizontal collapse, cracking, and fracture structures. Damage due to liquefaction can be characterized by objects and buildings being turned over, rotated, gone, sunk in water, or sunk in mud. Damage due to tsunamis is characterized by objects, buildings, or structures being

15   washed away from the shoreline by a water current.

Survey sites in the western coastal area of Palu Bay included Site 1 to 6. Site 1 (Donggala City) is located at the mouth of the bay. There are a fishing port and an inter-island port on this site. A survival fisherman told that he was on a ship when the tsunami struck. He saw tempestuous seawater not far from the position of the ships in the vicinity of the port of Donggala. The water propagated from the tempestuous seawater towards warfs in the ports caused a fishing boat rose to the dock floor.

20   Move south towards site 2 and 3 namely Loli Dondo Village and Loli Saluran Village. Both of these sites have the same characteristics, where many resident houses built on the right and left side of the Trans Sulawesi road. The part of the housing that is closer to the beach is mostly destroyed, while the part of the housing that is closer to the hill side has moderate damage.

Site 4 and 5 (Watusampu Village and Tipo Village) also have similar characteristics. Topography in the form of a steep

25   surface due to a row of hills on the west coast of Palu Bay. These hills are a source of sand for building materials. So there are a lot of sand mining activities at these two sites. In Watusampu site, measurement was carried out around the naval base of Indonesian Navy, where a navy patrol boat was lifted from its mooring site to the mainland. Approaching the tip of Palu Bay on the west side is the Site 6 (Silae) which is an urban area with a dense population. The main road on this site is very close (20-30 m) to the coastline. Settlement around the road was badly damaged. A 4-star hotel suffered serious structural

30   damage but did not collapse.

The sites in the southern coast of the bay consists of sites 7 to 9 (Lere, Besusu Barat, and Talise), lie in the end of Palu Bay, have a sloping topography, the highest population, the most fatalities, and the worst damage. In Besusu Barat, a steel bridge

with a span of 300 m was collapsed. The density of buildings in this area seems to have prevented the tsunami from reaching further inland. Witnesses who were on the banks of the Palu River during the earthquake and tsunami event said that the bridge collapsed during the earthquake and before the tsunami arrived. Amateur videos taken from the bridge abutment provide clues to the depth of flow. Measurements of trees and small buildings around the bridge indicate the depth of the tsunami flow around 3.25 m. Most of the victims came from this site because it is a densely populated area, with many offices and business activities as well as open public spaces. In addition, there was the Palu Nomoni festival, a public party that invited large crowds, at the time of the tsunami on Besusu and Talise beach and its surroundings.

Survey sites in the eastern coast area of Palu Bay consisted of Site 10 to 16. Site 10 was located in Tondo. This area has many private boarding houses for students of the University of Tadulako, the biggest university in the city of Palu. The topography of this area is relatively steep with a slope of 0.04 (4%). The runup height of 10.73 m is the highest in this survey (Fig. 5) caused a few building surviving. This area was very crowded when the earthquake and tsunami event. Most students were in their boarding houses during the earthquake because it occurred after working hours. Surprisingly, fewer than 10 deaths were recorded. This is likely due to most of the young residents have agility to save themselves when the tsunami arrived.

North of Tondo is Site 11 (Layana). This site is a trade complex that supports the economic activities of Palu City in particular and Central Sulawesi Province in general. The buildings damaged at this site functioned as shops, warehouses, and corporate offices.

Site 12 and 13 are Mamboro and Taipa. High flow depth of 5.36 m caused severe damage houses and casualties in Mamboro village. A stream covered fully by debris. In Taipa village, runup and flow depth reach 4.88 m and devastated passenger terminal, ferry crane, and navigation control building. Taipa is passenger port serves Sulawesi Island to others. Site 14 (Pantoloan) is the biggest port in the bay where containers floated off and port crane collapsed. North of Pantoloan port is Wani port (Site 15). Here, we found terrible damage especially houses of fisherman community, collapsed coastal structures, and a ship lifted to land. Runup, inundation and flow depth were significant at this site. Site 16 (Lero) is northernmost of survey site inside Palu Bay. This site is face-to-face with Site 1 (western coast) where are also lies in mouth of Palu bay. Small harbour and its facilities totally destroyed. The two last sites were Tanjung Padang and Lende. These site located outside of Palu Bay and close to the epicenter. Tsunamis were felt just like tide wave. They destroy a little part of housing and agricultural field. Coastal area between site 16 and 17 is steep slope of hilly area, very few houses and no tsunami impact found.

We made videos to document damage along Trans Sulawesi Road. Then those are compared to Google Street View® before the tsunami occurrence. From the videos, it can be seen that the severe damage was limited in 150 m from coastline. Impact of the tsunami towards structures and coastal environment summarized in Table 2.

A detail measurement was done to a special phenomenon. A reinforced concrete bridge with simple support beam type on Cumi-cumi Road, Palu City (Fig 6b) shifted as far as 9.7 m. It gives a clue regarding the tsunami's strength. This bridge is made of reinforced concrete with a bridge span of 5.0 m and a width of 19.1 m. It passed over an open channel which had a

width of 4.1 m and a depth of 1.6 m. It had 14 beam girders with dimensions of 0.25 m × 0.30 m with a girder distance of 1.35 m. Its plate had a thickness of 0.20 m. Based on these dimensions, the surface area of the bridge was 244.7 m$^2$, the volume was 23.4 m$^3$, and the mass was approximated to be around 56 tons. The bridge was estimated to have been submerged by tsunami water as deep as 2.5-4.0 m based on the tsunami marks around it. Debris caught in the bridge fence

5  (Fig. 6b) was evidence of the tsunami water soaking the bridge. The shift stopped because the bridge body was stuck in the wall of a building. Furthermore, we can investigate this case with the help of Google Earth, as shown in Fig. 7, where Figs. 7a and 7b show satellite images taken on September 26, 2017, and October 2, 2018, respectively. As shown, the asphalt layer of the road was broken and the bridge over the open channel was shifted away from the coast by the tsunami. The position of this bridge is at the end of Palu Bay (-0.88123°S; 119.83907°E).

10  **6 Coastal Landslides**

Total coastal landslides in Palu Bay related to 28 September 2018 event occurred at 7 locations (Sassa and Takagawa, 2018), 6 locations (Arikawa, 2018) or 10 locations (Omira et al., 2019). Our team found two locations of coastal landslides. These add landslide locations which have been found by other survey team. The two locations are around the river mouth in Donggala City (Figs. 11 and 12) and around the river mouth in Lero Village (Figs. 13 and 14). Landslides in Donggala were

15  indicated by the loss of land around the Donggala River. Around 30 houses were reported to have suddenly sunk along with some of the residents. The wharf in the port of Donggala dropped by about 80 cm. The pile that was being installed for the foundation of a large building sank deep into the soil layer suddenly and was no longer found.

In Lero village, some houses and their inhabitants drowned when the tremor struck. Figure 12 shows a house going down so that the ceiling was in the previous floor position. A typical house in Indonesia has a ceiling height of 3 to 4 m. This is an

20  indication that the landslide in Lero village has reduced the land surface in the range of 3 to 4 m. In addition, an eyewitness reported that seabed around 10 m from coastline changed from 1 m deep to "invisible" depth seen by naked eyes. He heard roaring sound a minute after the main earthquake.

**7 Conclusions**

This study reported the results of a post-tsunami field survey conducted after the 2018 Sulawesi Tsunami. The results show

25  that the runup heights reached 10.73 m in Tondo and the inundation distance was 488 m in Layana. Tondo area has a steep slope coast whereas Layana area has a flat topography. Flow depths were detected more than 2 m for sites which had significant damage. Tsunami events are concentrated in the bay, this indicates the type of local tsunami. Most informans in the survey site testified that there were three main tsunami waves that reached the coastal zone in Palu Bay. The second was the highest. The arrival time of waves varied according to location. It is around 3 minutes in Donggala City and Lero

30  Village, and around 10 minutes in the end of Palu Bay (Lere, Besusu Barat, and Talise) after the $M_W$ = 7.5 main earthquake event.

The tsunami waves that hit the coastal zone in Palu Bay were very strong as indicated by massive damage at each site we surveyed. The severe damage was limited within 150 m from coastline. These include the shifting of a 38-ton bridge. Coastal landslides detected by our team in Donggala City (lost surface area of 10,068 m$^2$) and Lero Village (lost suface area of 22,971 m$^2$) gives additional evidence towards coastal landslides found by other team as reported by Arikawa et al. (2018) and Omira et al. (2019). 
[revised manuscript text omitted]
 station were at Site 1 and Site 14. Coastal landslides detected were located at Site 1 and Site 16.

[Figure]

**Figure 2:** Evidence of tsunami runup and inundation. (a) Debris left behind in the residential area of Tondo, (b) debris caught in a tree in Mamboro, (c) and (d) debris stuck in a tree in Tondo, (e) leaves turned brown due to being submerged in salt water, (f) a tree had green leaves at the top and brown at the lower part, indicating the tsunami inundation height (flow depth) limit in Layana, (g) debris lodged on top of a building, (h) broken house element showing tsunami water level, (i) watermark on a house wall in Lero village, (j) sand deposit on building floor in Donggala City, (k) a 45-m-long ship moved to land in Wani harbour, (l) interview with an survivor in Mamboro.

[Figure]

**Figure 3:** Measurement results of inundation distances.

[Figure]

**Figure 4:** Measurement results of runup heights.

[Figure]

**Figure 5:** Transects of beach where tsunami wave arrived. The longest inundation distance is at the Layana site and the highest runup is at the Tondo site.

[Figure]

( a )  ( b )

**Figure 6:** (a) Damage caused by the tsunami in Tondo, a residential complex where a lot of private boarding houses were inhabited by students at the University of Tadulako, and (b) a reinforced concrete bridge on Cumi-cumi Road Palu City shifted by 9.7 m by the tsunami.

[Figure]

( a )                                    ( b )

**Figure 7:** Satellite images taken on (a) September 26, 2017 and (b) October 2, 2018, showing the bridge shift.

[Figure]

**Figure 8:** Water level recording at the Pantoloan tidal station managed by Geospatial Information Agency (Sudibyo, 2018).

[Figure]

**Figure 9:** Magnified view of Fig. 9, sourced Geospatial Information Agency (Sudibyo, 2018).

[Figure]

( a )                    ( b )

( c )                    ( d )

**Figure 10:** Landslide in Donggala City. (a) A trestle dropped 0.8 m in Donggala Port, (b) a building on the seaside slip down significantly, (c) the surface of an alley in a settlement dropped 0.4 m, and (d) a layered courtyard with paving blocks dropped around 1.5 m.

[Figure]

( a ) ( b )

Figure 11: Possible landslide areas in Donggala (yellow dotted lines). Images were obtained from Google Earth. Satellite images taken on(a) 6 July, 2016 (more than a year before the earthquake) and (b) 2 October, 2018 (4 days after the earthquake and tsunami). The yellow bounded area is around 10,068 m$^2$ or 1 hectare. Number 1, 2, 3 and 4 in Fig. 12 b corresponds to Fig. 11 (a), (b), (c), and (d).

[Figure]

Figure 12: Quick landsubsidence in Lero Village. Photograph taken two weeks after the event. Some houses dropped suddenly, around 3-4 m, when the earthquake occurred. Residents of these houses, especially that indicated by the oval, could not save themselves. The yellow dotted line is the former coastline.

[Figure]

**Figure 13:** Quick land subsidence in Lero Village. Satellite images taken on (a) 7 April, 2016, and (b) 2 October, 2018, from Google Earth, showing conditions after the earthquake and tsunami. The area of land that dropped is 22,971 m$^2$ or almost 2.3 hectares.

---

## Author Comment (AC2) · 19 Sep 2019

Referee 2 – Authors Interaction

We would like to thank Prof Ahmet Cevdet Yalciner for the constructive comments and suggestions towards improving our manuscript. Our response manuscript was uploaded in the form of a supplement. We summarize comments from Referee 2, author's response, and author's changes in manuscript as follows.

Comment 1: Page-1 Line 14: Indicating the name of the university mentioned would provide more clear information. Line 22-23: Do the authors have any reference for the earthquake parameters given? Line 23-24: Do the authors have any reference for the numbers reported?

[Figure]

Response 1: Page-1 Line 14 The name of the university is Tadulako University, the biggest university in Palu City. We remove it from abstract and add it in paragraph. Line 22-23 The earthquake parameters given are from Meteorological, Climatological and Geophysical Agency (Indonesian: Badan Meteorologi, Klimatologi, dan Geofisika, BMKG) (http://inatews.bmkg.go.id/?act=tsuevents and https://www.bmkg.go.id/press-release/?p=gempabumi-tektonik-m7-7-kabupaten-donggala-sulawesi-tengah-pada-hari-jumat-28-september-2018-berpotensi-tsunami&tag=press-release&lang=ID      ). The parameters are also available in BMKG's earthquake catalog. Line 23-24 The numbers reported are from the Indonesian National Disaster Management Agency (Indonesian: Badan Nasional Penanggulangan Bencana, BNPB) which was broadcasted by media. We took one of them from https://nasional.tempo.co/read/1138400/jumlah-korban-tewas-terkini-gempa-dan-tsunami-palu-2-113-orang/full&view=ok      We      add these references in the manuscript.

Changes in manuscript: P1 L13 supplement ..... The survey results show that the runup height and inundation distance reached 10.7 m in Tondo and 488 m in Layana respectively. .....

P1 L17-19 supplement ..... On Friday, September 28, 2018, at 18:02:44 Central Indonesia Time (UTC + 8), Palu Bay was hit by a strong earthquake with magnitude MW = 7.5. The epicenter was located at -0.22°N 119.85°E at a depth of 10 km and 27 km northeast of Donggala City (BMKG, 2018). .....

P1 L19-20 supplement ..... As of October 21, 2018, as many as 2,113 people were killed, 1,309 missing, and 4,612 injured (Hadi and Kurniawati, 2018). .....

Comment 2: Page-3 Line 2-5: "For tsunamis, post-incident surveys are often carried out. Major tsunamis such as: : :." Only stating some of the tsunami post-event surveys like "giving examples" may not be appropriate. Two suggestions: Either state the importance and relation of them with this study OR delete these sentences. Line 9: "Observation of damage was also conducted." Too general sentence. What kind of

damage data is collected? Any details on the data collection processes?

Response 2: Page-3 Line 2-5 We delete these sentences "For tsunamis, post-incident surveys are often carried out. Major tsunamis such as: : :." Line 9 We emphasized damage to buildings and infrastructures. We identified the difference about damage by earthquake, liquefaction and tsunami. We made videos to document damage along Trans Sulawesi Road, compared them to Google Street View$^{®}$ , Google Map and Google Earth and concluded that the severe damage was limited in 150 m from coastline. We also measured dimension of a bridge because we assume it was special case we found. We add about this in paragraphs of the manuscript.

Changes in manuscript 2: P2 L25-28 supplement ..... Ulrich et al. (2019) assume that a source related to earthquake displacements is probable and that landsliding may not have been the primary source of the tsunami. On the contrary, Takagi et al. (2019), Sassa and Takagawa (2018), Arikawa et al. (2018) tend to assume that landslides produced the tsunami. Field surveys play an important role to support seeking answer for question arisen. .....

P2 L31-31 supplement ...... Tsunami flow depth on land was also measured in some sites. In addition, tsunami arrival time was analyzed and observation of buildings and infrastructures damage was conducted. .....

P5 L4-8 supplement ..... Damage observation was carried out at each site of surveys. We emphasized on damage to buildings and infrastructures although other kind of damage are interesting, such as vegetation, shoreline, and properties (cars, boats, fisherman tools, etc.). Videos and photographs were produced to assess the damage. Video recorded along trans Sulawesi Road were compared to Google Street View, Google Map, and Google Earth in order to assess the distance of damage from coastline. In addition, detail measurement of dimension was done for special object (for instance bridge) which is useful for tsunami force analysis. .....

Comment 3: Page 9: Line 14: "There were three main tsunami waves that reached the

beach." Which beach? Not clear. Line 14: "The first wave was relatively low." With respect to what? You should State it more clearly and in an understandable way. Line 18: "The wave that hit the beach was quite high." This sentence by itself does not provide any meaningful information.

Response 3: Page-9 Line-14 We modify "There were three main tsunami waves that reached the beach." to be "Most informants in the survey site testified that there were three main tsunami waves that reached the coastal zone inside Palu Bay. The second was the highest". Line-18 We replace "The wave that hit the beach was quite high" with "The tsunami waves that hit the coastal zone in Palu Bay were very strong as indicated by massive damage at each site we surveyed. These include the shifting of a 38-ton bridge."

Changes in manuscript 3: P9 L27-29 supplement ..... Most informants in the survey site testified that there were three main tsunami waves that reached the coastal zone inside Palu Bay. The second was the highest. .....

P10 L1-2 supplement ..... The tsunami waves that hit the coastal zone in Palu Bay were very strong as indicated by massive damage at each site we surveyed. The severe damage was limited within 150 m from coastline. .....

Comment 4: General Comments: - A brief summary and citation of previously published papers on 2018 Palu Event field survey is necessary. - Section 5.1, Aftershock information is not related with the focus of this study and the work done. - The conclusion section should be rewritten by clear sentences and providing a comprehensive summary of the results obtained. For example, Giving ranges such as "2 to10 m and the inundation distances were 80 to 500 m." Or "The arrival time of waves varied from 3 to 10 minutes.." does not provide satisfying information. The authors, at least, may add the locations of these measurements.

Response 4: Thanks for the suggestions. We provide a brief summary and citation of previously published papers e.g. Omira et al (2019), Mikami et al (2019), Yalciner

et al. (2018), Muhari et al (2018), and Arikawa et al. (2018). We remove section 5.1 about aftershock information which is also suggested by Referee 1. We modify the sentences, the ranges of numbers are replaced by the maximum and significant run up height and inundation distance with mentioning the locations. Your comments and suggestions are accommodated in uploaded supplement file (and a revised paper manuscript if this process continue to next phase).

Changes in manuscript 4: P3 L5-12 supplement Many groups have conducted field surveys of the Sulawesi tsunami event or also known as Palu tsunami. Muhari et al. (2018) investigated wave height and inundation depth at several points with a focus around the end of the bay. A UNESCO international tsunami survey team surveyed 125 km of coastline along the Palu Bay up to the earthquake epicentre region. The team performed 78 tsunami runup and inundation height measurements throughout the surveyed coastline (Omira et al., 2019; Yalciner et al., 2018). Putra et al (2019) focus more on tsunami deposits. Meanwhile, Arikawa et al. (2018), Sassa and Takagawa (2018), and Takagi et al. (2019) each conducted a survey related to coastal subsidence, coastal liquefaction or submarine landslide detected in Palu Bay. This survey data can be combined with data from other groups, especially we contribute to provide data of runup, inundation distance, and damage.

Please also note the supplement to this comment:
https://www.nat-hazards-earth-syst-sci-discuss.net/nhess-2019-91/nhess-2019-91-AC2-supplement.pdf

**Supplement:**

[revised manuscript text omitted]

model, i.e. Boussinesq model and a nonlinear shallow water wave model. Yalciner (2001) conducted field survey and modeling of the 1999 Izmit tsunami which the location had similarity in geographycal feature, earthquake magnitude and tsunami mechanism with recent Sulawesi case. More broadly, these data can be used for disaster mitigation and rebuilding of the affected areas by the 2018 Sulawesi Tsunami.

5   Many groups have carried out field surveys of the Sulawesi tsunami event or also known as Palu tsunami. Muhari et al. (2018) investigated wave height and inundation depth at several points with a focus around the end of the bay. A UNESCO international tsunami survey team surveyed 125 km of coastline along the Palu Bay up to the earthquake epicentre region. The team performed 78 tsunami runup and inundation height measurements throughout the surveyed coastline (Omira et al., 2019; Yalciner et al., 2018). Putra et al (2019) focus more on tsunami deposits. Meanwhile, Arikawa et al. (2018), Sassa and
10  Takagawa (2018), and Takagi et al. (2019) each conducted a survey related to coastal subsidence, coastal liquefaction or submarine landslide detected in Palu Bay. This survey data can be combined with data from other groups, especially we contribute to provide data of runup height, inundation distance, flow depth, and damage at different points and coordinates.

**2 Survey Details**

A team from National Cheng Kung University, Taiwan, and Universitas Jenderal Soedirman, Indonesia, arrived at Sis Aljufri
15  Airport in Palu City at 06:00 a.m. Central Indonesia Time on October 11, 2018, thirteen days after the tsunami event. Studies have shown that surveys can be carried out successfully within two to three weeks of an event (Synolakis and Okal, 2005). Starting from the afternoon of October 11, a field survey was conducted until October 19 evening, for a survey period of 9 days. The emergency response period for the disaster area was determined by the Indonesian government to be one month (September 28 to October 26, 2018). The victim evacuation period was two weeks (September 28 to October 12). This
20  means that the survey was conducted in the emergency response stage, one day before the victim evacuation period ended. During this period, cleaning of area impacted by tsunami was still in progress, so that debris could be seen in the disaster area.

[revised manuscript text omitted]

Runup heights were corrected to calculate heights above sea level at the time of survey by using WXTide software version 4.7, available at www.wxtide32.com/index.html. We use Donggala station, the closest station listed in the software, for correcting and assume no significant variations on the sea level inside Palu Bay.

Damage observation was carried out at each site of surveys. We emphasized on damage to buildings and infrastructures although other kind of damage are interesting, such as vegetation, shoreline, and properties (cars, boats, fisherman tools, etc.). Videos and photographs were produced to assess the damage. Video recorded along trans Sulawesi Road were compared to Google Street View, Google Map, and Google Earth in order to assess the distance of damage from coastine. In addition, detail measurement of dimension was done for special object (for instance bridge) which is useful for tsunami force analysis.

**3 Inundation and Runup Measurements Result**

Runup is the maximum ground elevation wetted by the tsunami on a sloping shoreline. In the simplest case, the runup value is recorded at maximum horizontal inundation distance (IOC Manuals and Guides No. 37, 2014). The measurement results are shown in Table 1 and Figs. 3-5. The measurement values in the table has been corrected with tides. Runup height and inundation distance vary from site to site.

Western coast of Palu Bay comprises of Site 1 to 6. Site 1 (Donggala City) is located at the mouth of the bay. Runup height and inundation distance at this site were not significant. Site 2 and 3 namely Loli Dondo Village and Loli Saluran Village where had runup height for both sites is relatively the same, 2.53 m and 2.18. Inundation distance were short due to steep hills towards the mainland. Site 4 and 5 (Watusampu Village and Tipo Village) had height of runup, 6.63 m and 7.79 m. The inundation distances were 71.51 m and 91.11 m. High runup with short inundation was influenced by steep topography. The highest runup for western coast was found in Tipo (7.79 m), followed by Watusampu (6.63 m).

The site at the southern coast of the bay (end of Palu Bay) consists of sites 7 to 9 (Lere, Besusu Barat, and Talise). The runup heights at these site were low at 1.40 m and 1.12 m for Lere and Besusu Barat. Talise had a higher runup of 3.02 m, but all the three had almost the same inundation distance between 200 to 250 m. The density of buildings in this area seems to have prevented the tsunami from reaching further inland. Flat topography caused runup elevation that did not differ much from sea levels.

Survey sites in the eastern coast area of Palu Bay consists of Site 10 to 16. Site 10 was located in Tondo. The topography of this area is relatively steep with a slope of 0.06 (6%). Evidence of tsunami water rise was in the form of debris on top of buildings, truncated building elements, collapsed walls, trash carried away, and fixed debris. A survivor also showed the highest places of tsunami water in this area. A total of 4 cross sections of these coast were measured by our team. The measured runup heights were 10.73, 7.97, 10.14, and 8.50 m, respectively, as shown in Table 1. The runup height of 10.73 m is the highest in this survey (Fig. 5) caused a few building surviving. Omira et al. (2019) shows that the highest runup from

their field survey was in Benteng Village with height of 9.1 m. Benteng Village (in western coast) is viz-a-viz with highest runup location of our survey in Tondo (in eastern coast).

North of Tondo is Site 11 (Layana). The topography of this site is relatively flat with a slope of 0.013 (1.3%). Because of this sloping topography, the tsunami wave reached as far as 488 m inland. It was the longest distance among others. The runup points reached 6.57 m and 2.78 m in this site. Both varied greatly because many buildings have long and wide walls that stemmed the tsunami flow further inland.

Site 12 and 13 are Mamboro and Taipa. Runup height of 3.5 m and flow depth of 5.36 m caused severe damage houses and casualties in Mamboro. In Taipa, runup of 4.88 m reached the roof of passenger terminal of Taipa port. North of Pantoloan port is Wani port (Site 15). Runup, inundation and flow depth were significant at this site, 3.58 m and 5.12 m respectively. Site 16 (Lero) is northernmost of survey site inside Palu Bay. This site is face-to-face with Site 1 where are also lies in mouth of Palu bay. The two last sites were Tanjung Padang and Lende. These site located outside of Palu Bay and close to the epicenter. Around 1 m runup hit the both site. Coastal area between site 16 and 17 is steep slope of hilly area and no tsunami footage existed.

**4 Tsunami Arrival Time**

Arrival time of a tsunami wave is one of the main parameters calculated in tsunami modeling. The time needed for the tsunami wave to propagate from earthquake source location to the coast is defined by the estimated time of arrival (ETA) (Strunz et al., 2011). It is important related to early warning system.

Tidal records may provide a clue on tsunami arrival time. The tidal station closest to the disaster site is Pantoloan Tidal Station. This station is located inside Palu Bay, on a pier in Pantoloan Port and operated by the Agency of Geospatial Information. When the earthquake and tsunami occurred, the recording equipment was not damaged but the data transfer stopped because the communication network was interrupted. Fig. 8 and Fig. 9 shows the water level recorded when the tsunami arrived. The maximum low tide (6.74 m) was at 18:08 local time and the maximum tide (10.55 m) was at 18:10 local time. This means that the tsunami wave height recorded at the station was 3.8 m. This wave height can be seen more clearly in Fig. 9, which is from the same source as that for Fig. 8. In addition, the first tsunami wave arrived at 18:07, with the wave trough at 18:08 and the wave crest at 18:10 local time (UTC+8).

Other hint regarding tsunami arrival time are based on videos on social media, internet, and television, as well as eyewitnesses. More than one tsunami wave hit the coastal zone in Palu Bay. Most witnesses stated that three tsunami waves had arrived. The first was less than 1-m high. The second and third waves were much higher, and were quantified by measurements in this survey. The number of tsunami waves and their height order were similar to the 17 July 2006 Tsunami in Java. That event also had three tsunami waves which the first one was of little magnitude and was followed by the second wave which was the highest one (Lavigne et al., 2007). Witnesses did not give an exact arrival time of the tsunami wave on

the coastal zone in Palu Bay. Generally, they referred to prayer times as a guide. Indonesia is majority Muslim. The time of the earthquake and tsunami is close to one of the Muslim worship times in the afternoon, which coincides with a sunset called "maghrib" prayer. The prayer schedule circulated by the Ministry of Religion of the Republic of Indonesia for the area of Palu City and Donggala Regency indicates that the starting time of "maghrib" prayer period on September 28, 2018, was 17:58 local time. Normally, there are two call sounded from a mosque as starting time sign for praying. The first call is called "adzan" and the second call is called "iqamah". The period between the two call is 10 minutes. Some news, videos, and witnesses show that the tsunami came when people were preparing to pray, between "adzan" and "iqamah". The $M_W = 7.5$ earthquake occurred at 18:04. This shows that the tsunami waves came less than 10 minutes after the earthquake or between 18:05 and 18:15 local time, different for each site in the disaster area. Thus, the testimony of the witnesses was consistent with the detection of tidal gauges at the Pantoloan station. The important note from the September 2018 event is that the tsunami arrival time was very short.

**5 Buildings and Infrastructures Damage**

We identified damages to buildings and structures caused by the Sulawesi event 2018 into 3 types, namely damage due to earthquakes, liquefaction, and tsunamis. Damage caused by earthquakes is characterized by horizontal collapse, cracking, and fracture structures. Damage due to liquefaction can be characterized by objects and buildings being turned over, rotated, gone, sunk in water, or sunk in mud. Damage due to tsunamis is characterized by objects, buildings, or structures being washed away from the shoreline by a water current.

Survey sites in the western coastal area of Palu Bay included Site 1 to 6. Site 1 (Donggala City) is located at the mouth of the bay. There are a fishing port and an inter-island port on this site. A survival fisherman told that he was on a ship when the tsunami struck. He saw tempestuous seawater not far from the position of the ships in the vicinity of the port of Donggala. The water propagated from the tempestuous seawater towards warfs in the ports caused a fishing boat rose to the dock floor.

Move south towards site 2 and 3 namely Loli Dondo Village and Loli Saluran Village. Both of these sites have the same characteristics, where many resident houses built on the right and left side of the Trans Sulawesi road. The part of the housing that is closer to the beach is mostly destroyed, while the part of the housing that is closer to the hill side has moderate damage.

Site 4 and 5 (Watusampu Village and Tipo Village) also have similar characteristics. Topography in the form of a steep surface due to a row of hills on the west coast of Palu Bay. These hills are a source of sand for building materials. So there are a lot of sand mining activities at these two sites. In Watusampu site, measurement was carried out around the naval base of Indonesian Navy, where a navy patrol boat was lifted from its mooring site to the mainland. Approaching the tip of Palu Bay on the west side is the Site 6 (Silae) which is an urban area with a dense population. The main road on this site is very close (20-30 m) to the coastline. Settlement around the road was badly damaged. A 4-star hotel suffered serious structural damage but did not collapse.

The sites in the southern coast of the bay consists of sites 7 to 9 (Lere, Besusu Barat, and Talise), lie in the end of Palu Bay, have a sloping topography, the highest population, the most fatalities, and the worst damage. In Besusu Barat, a steel bridge with a span of 300 m was collapsed. The density of buildings in this area seems to have prevented the tsunami from reaching further inland. Witnesses who were on the banks of the Palu River during the earthquake and tsunami event said that the

5      bridge collapsed during the earthquake and before the tsunami arrived. Amateur videos taken from the bridge abutment provide clues to the depth of flow. Measurements of trees and small buildings around the bridge indicate the depth of the tsunami flow around 3.25 m. Most of the victims came from this site because it is a densely populated area, with many offices and business activities as well as open public spaces. In addition, there was the Palu Nomoni festival, a public party that invited large crowds, at the time of the tsunami on Besusu and Talise beach and its surroundings.

10     Survey sites in the eastern coast area of Palu Bay consisted of Site 10 to 16. Site 10 was located in Tondo. This area has many private boarding houses for students of the University of Tadulako, the biggest university in the city of Palu. The topography of this area is relatively steep with a slope of 0.04 (4%). The runup height of 10.73 m is the highest in this survey (Fig. 5) caused a few building surviving. This area was very crowded when the earthquake and tsunami event. Most students were in their boarding houses during the earthquake because it occurred after working hours. Surprisingly, fewer than 10

15     deaths were recorded. This is likely due to most of the young residents have agility to save themselves when the tsunami arrived.

North of Tondo is Site 11 (Layana). This site is a trade complex that supports the economic activities of Palu City in particular and Central Sulawesi Province in general. The buildings damaged at this site functioned as shops, warehouses, and corporate offices.

20     Site 12 and 13 are Mamboro and Taipa. High flow depth of 5.36 m caused severe damage houses and casualties in Mamboro village. A stream covered fully by debris. In Taipa village, runup and flow depth reach 4.88 m and devastated passenger terminal, ferry crane, and navigation control building. Taipa is passenger port serves Sulawesi Island to others. Site 14 (Pantoloan) is the biggest port in the bay where containers floated off and port crane collapsed. North of Pantoloan port is Wani port (Site 15). Here, we found terrible damage especially houses of fisherman community, collapsed coastal structures,

25     and a ship lifted to land. Runup, inundation and flow depth were significant at this site. Site 16 (Lero) is northernmost of survey site inside Palu Bay. This site is face-to-face with Site 1 (western coast) where are also lies in mouth of Palu bay. Small harbour and its facilities totally destroyed. The two last sites were Tanjung Padang and Lende. These site located outside of Palu Bay and close to the epicenter. Tsunamis were felt just like tide wave. They destroy a little part of housing and agricultural field. Coastal area between site 16 and 17 is steep slope of hilly area, very few houses and no tsunami

30     impact found.

We made videos to document damage along Trans Sulawesi Road. Then those are compared to Google Street View® before the tsunami occurrence. From the videos, it can be seen that the severe damage was limited in 150 m from coastline. Impact of the tsunami towards structures and coastal environment summarized in Table 2.

A detail measurement was done to a special phenomenon. A reinforced concrete bridge with simple support beam type on Cumi-cumi Road, Palu City (Fig 6b) shifted as far as 9.7 m. It gives a clue regarding the tsunami's strength. This bridge is made of reinforced concrete with a bridge span of 5.0 m and a width of 19.1 m. It passed over an open channel which had a width of 4.1 m and a depth of 1.6 m. It had 14 beam girders with dimensions of 0.25 m × 0.30 m with a girder distance of 1.35 m. Its plate had a thickness of 0.20 m. Based on these dimensions, the surface area of the bridge was 244.7 m$^2$, the volume was 23.4 m$^3$, and the mass was approximated to be around 56 tons. The bridge was estimated to have been submerged by tsunami water as deep as 2.5-4.0 m based on the tsunami marks around it. Debris caught in the bridge fence (Fig. 6b) was evidence of the tsunami water soaking the bridge. The shift stopped because the bridge body was stuck in the wall of a building. Furthermore, we can investigate this case with the help of Google Earth, as shown in Fig. 7, where Figs. 7a and 7b show satellite images taken on September 26, 2017, and October 2, 2018, respectively. As shown, the asphalt layer of the road was broken and the bridge over the open channel was shifted away from the coast by the tsunami. The position of this bridge is at the end of Palu Bay (-0.88123°S; 119.83907°E).

**6 Coastal Landslides**

Total coastal landslides in Palu Bay related to 28 September 2018 event occurred at 7 locations (Sassa and Takagawa, 2018), 6 locations (Arikawa, 2018) or 10 locations (Omira et al., 2019). Our team found two locations of coastal landslides. These add landslide locations which have been found by other survey team. The two locations are around the river mouth in Donggala City (Figs. 11 and 12) and around the river mouth in Lero Village (Figs. 13 and 14). Landslides in Donggala were indicated by the loss of land around the Donggala River. Around 30 houses were reported to have suddenly sunk along with some of the residents. The wharf in the port of Donggala dropped by about 80 cm. The pile that was being installed for the foundation of a large building sank deep into the soil layer suddenly and was no longer found.

In Lero village, some houses and their inhabitants drowned when the tremor struck. Figure 12 shows a house going down so that the ceiling was in the previous floor position. A typical house in Indonesia has a ceiling height of 3 to 4 m. This is an indication that the landslide in Lero village has reduced the land surface in the range of 3 to 4 m. In addition, an eyewitness reported that seabed around 10 m from coastline changed from 1 m deep to "invisible" depth seen by naked eyes. He heard roaring sound a minute after the main earthquake.

**7 Conclusions**

This study reported the results of a post-tsunami field survey conducted after the 2018 Sulawesi Tsunami. The results show that the runup heights reached 10.73 m in Tondo and the inundation distance was 488 m in Layana. Tondo area has a steep slope coast whereas Layana area has a flat topography. Flow depths were detected more than 2 m for sites which had significant damage. Tsunami events are concentrated in the bay, this indicates the type of local tsunami. Most informans in the survey site testified that there were three main tsunami waves that reached the coastal zone in Palu Bay. The second was

the highest. The arrival time of waves varied according to location. It is around 3 minutes in Donggala City and Lero Village, and around 10 minutes in the end of Palu Bay (Lere, Besusu Barat, and Talise) after the $M_W = 7.5$ main earthquake event.

The tsunami waves that hit the coastal zone in Palu Bay were very strong as indicated by massive damage at each site we
5    surveyed. The severe damage was limited within 150 m from coastline. These include the shifting of a 38-ton bridge. Coastal landslides detected by our team in Donggala City (lost surface area of 10,068 $m^2$) and Lero Village (lost suface area of 22,971 $m^2$) gives additional evidence towards coastal landslides found by other team as reported by Arikawa et al. (2018) and Omira et al. (2019). 
[revised manuscript text omitted]
 station were at Site 1 and Site 14. Coastal landslides detected were located at Site 1 and Site 16.

[Figure]

**Figure 2:** Evidence of tsunami runup and inundation. (a) Debris left behind in the residential area of Tondo, (b) debris caught in a tree in Mamboro, (c) and (d) debris stuck in a tree in Tondo, (e) leaves turned brown due to being submerged in salt water, (f) a tree had green leaves at the top and brown at the lower part, indicating the tsunami inundation height (flow depth) limit in Layana, (g) debris lodged on top of a building, (h) broken house element showing tsunami water level, (i) watermark on a house wall in Lero village, (j) sand deposit on building floor in Donggala City, (k) a 45-m-long ship moved to land in Wani harbour, (l) interview with an survivor in Mamboro.

[Figure]

**Figure 3:** Measurement results of inundation distances.

[Figure]

**Figure 4:** Measurement results of runup heights.

[Figure]

**Figure 5:** Transects of beach where tsunami wave arrived. The longest inundation distance is at the Layana site and the highest runup is at the Tondo site.

| | |
|---|---|
| ( a ) | ( b ) |
| (c) | (d) |

**Figure 6:** (a) Damage caused by the tsunami in Tondo, a residential complex where a lot of private boarding houses were inhabited by students at the University of Tadulako, and (b) a reinforced concrete bridge on Cumi-cumi Road Palu City shifted by 9.7 m by the tsunami. (c) Mamboro village with 90% houses destruction  (d) Asphalt layer of small road turned 90° in Tondo

[Figure]

5                                            ( a )                                                                           ( b )

**Figure 7:** Satellite images taken on (a) September 26, 2017 and (b) October 2, 2018, showing the bridge shift.

[Figure]

**Figure 8:** Water level recording at the Pantoloan tidal station managed by Geospatial Information Agency (Sudibyo, 2018).

[Figure]

**Figure 9:** Magnified view of Fig. 9, sourced Geospatial Information Agency (Sudibyo, 2018).

[Figure]

( a )                                                        ( b )

( c )                                                        ( d )

**Figure 10:** Landslide in Donggala City. (a) A trestle dropped 0.8 m in Donggala Port, (b) a building on the seaside slip down significantly, (c) the surface of an alley in a settlement dropped 0.4 m, and (d) a layered courtyard with paving blocks dropped around 1.5 m.

[Figure]

( a )                                                    ( b )

5   **Figure 11:** Possible landslide areas in Donggala (yellow dotted lines). Images were obtained from Google Earth. Satellite images taken on(a) 6 July, 2016 (more than a year before the earthquake) and (b) 2 October, 2018 (4 days after the earthquake and tsunami). The yellow bounded area is around 10,068 m$^2$ or 1 hectare. Number 1, 2, 3 and 4 in Fig. 12 b corresponds to Fig. 11 (a), (b), (c), and (d).

[Figure]

10   **Figure 12:** Quick landsubsidence in Lero Village. Photograph taken two weeks after the event. Some houses dropped suddenly, around 3-4 m, when the earthquake occurred. Residents of these houses, especially that indicated by the oval, could not save themselves. The yellow dotted line is the former coastline.

[Figure]

**Figure 13:** Quick land subsidence in Lero Village. Satellite images taken on (a) 7 April, 2016, and (b) 2 October, 2018, from Google Earth, showing conditions after the earthquake and tsunami. The area of land that dropped is 22,971 m$^2$ or almost 2.3 hectares.

---

## Author Comment (AC3) · 19 Sep 2019

Thank yo so much Sanidhya Nika Purnomo. We plotted the epicenter coordinate in the maps. You will see improved maps and pictures on a final manuscript if this process continues to next stage.

---

## Author Response (AR1)

**Authors Response**

**Referee 1 – Anonymous**

We would like to thank Referee 1 for the constructive comments and suggestions towards improving our manuscript.

We summarize comments from Referee 1, author's response, and author's changes in manuscript as follows. At the end of this authors response, we provide a manuscript showing highlights to mark referee comments. Then follow by a marked-up manuscript showing highlight to mark modification by author. Final form without highlight is available as revised manuscript.

**Comment 1:**

Overall This is an interesting paper that describes the results of field work after the2018 Sulawesi tsunami. The paper follows the general pattern of field work papers, and is important that such events are properly documents, so that modellers can then attempt to reproduce them. However, the paper suffers from lower than expected writing quality. The English is ok in some places, and poor in others. Also, the authors often repeat themselves. The more serious problem, however, comes from its unclear focus. Most of the paper deals with the tsunami damage, but at times the authors randomly include other information relating to aftershocks or landslides that are not related to the tsunami. Thus, several parts of the paper should be deleted, and the message should become more focused. Instead, the authors might want to describe the mechanisms of tsunami damage in more detail at each location (currently they only superficially describe some locations).

**Response 1:**

Thanks for very detailed and constructive comments from Referee1. We are improving our writing quality, including English. An native at an English proofreading service center would handle our manuscript. That's right we often repeated ourselves (23 times), now we already reduced it. We modify the writing in order to more focus by removing several parts you suggest, e.g aftershock and earth-surface landslide. Nevertheless, we preserve part about coastal landslides since we feel to have contribution on it. We strongly agree with your advice on description of the mechanism of tsunami damage. So that we add description on runup, inundation, and damage at each site in our text.

**MAJOR COMMENTS**

**Comment 2:**

By now a number of other field work papers have been published. Please find these, and cite them. Also, please explain what differences there are between your work and other papers.

**Response 2:**

Alright. We are adding other field work papers, i.e Omira et al (2019), Mikami et al. (2019), Muhari et al. (2018), Yalciner et al. (2018), Putra et al. (2019), Sassa & Takagawa (2019), Takagi (2019), Arikawa et al. (2018). The last three papers focus on coastal landslides, while Putra et al. evaluate runup based on tsunami deposit. Muhari et al. was probably the first team coming in disaster area. They gave early report as direction for other team coming later. Their results are preliminary and limited around Palu City, in the end of Palu Bay. The most close topic with us is by Omira et al. (UNESCO international team) and Mikami et al.. They measured more points than ours. Our several points intersect with their points. Nevertheless, they did not measure inundation distance as done by our team. We cite them and explain it in the manuscript.

**Change in manuscript 2:**

**P3 L6-15 marked-up/revised manuscript**

Many groups have carried out field surveys of the Sulawesi Tsunami event, also known as the Palu Tsunami. Muhari et al. (2018) investigated the wave height and inundation depth at several points with a focus around the end of the bay. A UNESCO international tsunami survey team surveyed 125 km of coastline along Palu Bay up to the earthquake epicentre region. The team performed 78 tsunami runup and inundation height measurements throughout the surveyed coastline (Omira et al., 2019; Yalciner et al., 2018). Mikami et al. (2019) measured runup height and inundation depth of 22 places and disscussed damage to coastal communities around Palu Bay. Putra et al. (2019) focused on tsunami deposits. Arikawa et al. (2018), Sassa and Takagawa (2018), and Takagi et al. (2019) each conducted a survey related to coastal subsidence, coastal liquefaction, or submarine landslides detected in Palu Bay. These survey data can be combined with data from other groups. In this study, we provide data of runup height, inundation distance, flow depth/inundation depth, and damage at different points and coordinates.

**Comment 3:**

P3, L24. In what way did the authors do this? How can they choose one point that can be representative for a tsunami that was as complicated as the one in this case?

**Response 3:**

P3 L24 version 1 manuscript. We modify the sentence. We mean that we chose some sites (not only one point) which had significant impact caused by the tsunami. We measured runup, inundation, and flow depth at 18 sites. On the first day of our survey we recorded situations along the Palu Bay. From these recordings we can roughly estimate sites with high runup and the long inundation. These points usually also have a severe level of damage. Besides, important places such as ports and densely populated areas are our priority. Measuring a coastal cross section at each certain distance, for example, every 1 km along 70 km of Palu Bay, might provide more representative data, but we have difficulty doing that mainly because it will take a long time. In addition, the areas affected by the tsunamis were also fragmented, not connected.

**Change in manuscript 3:**

**P3 L28-29 marked-up/revised manuscript**

..... 3) choosing sites that were significantly impacted by the tsunami; .....

**Comment 4:**

The English in the paper needs to be improved. In places the sentences are correct, and in others they are pretty poor.

**Response 4:**

Thanks for the assessment. We are trying to meet referee's suggestion, improving English in the manuscript thoroughly, and it was be checked by English proofreading service center.

**Comment 5:**

P4 L4-5 what kind of camera was used? Did the authors obtain a 360 degree view? Otherwise, in what way is this similar? Isthis going to be opened to other researchers? (if not, what is the point of writing this?)

**Response 5:**

P4\_L4-5 version 1 manuscript We delete "This method is similar to that used by Google Street View®, but we used simpler equipment." We do not have adequate reason to claim similar with

Google Street View. Only the idea may be similar with Google Street View® but the method and camera used was different. However, we think our video collections are useful. We plan to put them in the supplement in order to be opened to other researchers. We use Google Street View® for comparing with our videos to evaluate damage along Trans Sulawesi Road.

**Changes in manuscript 6:**

**P4 L10-11 marked-up/revised manuscript**

..... A camera on a moving car was operated to record the situation around the beach area. It produced a number of videos describing the damage (contained in the supplement).

**P5 L10-11 marked-up/revised manuscript**

.....Video recorded along trans Sulawesi Road were compared to Google Street View, Google Map, and Google earth in order to assess the distance of damage from coastine. ....

**Comment 6:**

P4 L19 and onwards. What is the point of talking so much about the rain, if the authors then dismiss the importance of it?

**Response 6:**

P4\_L19 version 1 manuscript We replace "Fortunately, from the point of view of conducting a survey, surface runoff due to rain seems insignificant and does not erase the tsunami footprint." with "It was a challenging work to look for tsunami footprint on surfaces that were exposed to surface runoff." In additon, we also shorten the paragraph containing about rain.

**Change in manuscript 6:**

**P4 L18-22 marked-up/revised manuscript**

..... October is the beginning of the rainy season in Indonesia, including Sulawesi. Palu City is located near the equator, as shown in Table 1. It is one of the driest areas in Indonesia, with rainfall recorded at the Mutiara Meteorology Station in 2017 of 774.3 mm. From the earthquake incident until the end of the survey, it rained four times, three of which occurred during our survey period, with a duration of less than 2 hours and with low to moderate intensity. It was challenging to find tsunami footprints on surfaces exposed to surface runoff caused by rain. .....

**Comment 7:**

P5 L2. Where all these measurements corrected for tide? Using which software? Are the datasets given in this paper those corrected for tide, or the original measurements? Also P5 L7-9, the location of these tidal stations needs to be shown in some figure. See also P5 L22, which indicates both corrected and uncorrected, making it unclear what the other numbers in the paper actually are.

**Response 7:**

P5 L2 version 1 manuscript Runup heights were corrected to calculate heights above sea level at the time of survey by using WXTide software version 4.7, available at <a href="http://www.wxtide32.com/index.html">www.wxtide32.com/index.html</a>. We used Donggala station listed in the software for correcting and assume no significant variations on the sea level inside Palu Bay. We modified that all number (runup and inundation) shown in the paper are corrected for tide. Thanks for rigorous comments.

P5\_L7-9 version 1 manuscript OK, we show Donggala and Pantoloan tidal stations in Fig. 1. We remove Mamuju station from the text.

P5\_L22 version 1 manuscript We modified that all number (runup, inundation, water elevation) shown in the paper are corrected for tide.

**Changes in manuscript 7:**

**P5 L5-7 marked-up/revised manuscript**

..... Runup heights were corrected to calculate heights above sea level at the time of the survey using WXTide software version 4.7, available at www.wxtide32.com/index.html. We used Donggala station, the closest station listed in the software, for corrections and assumed no significant variations in sea level inside Palu Bay.

**P5 L15-17 marked-up/revised manuscript**

..... The measurement results are shown in Table 1 and Figs. 3-5. The measurement values in the table are corrected based on the tides. Runup height and inundation distance vary from site to site.

**P5 L32 - P6 L1-2 marked-up/revised manuscript**

..... The measured runup heights were 10.73, 7.97, 10.14, and 8.50 m, respectively, as shown in Table 1. The runup height of 10.73 m is the highest in this survey (Fig. 5) caused a few building surviving. .....

**Comment 8:**

P6 L16. What is the point of this section? You are talking about earthquake damage, but this paper up to now is mostly about tsunami damage. Hence, it feels rather odd. I suggest just focusing on the damage by the tsunami, and delete this section.

P6 L24 If the bridge was shifted, it was damaged. Not sure what the authors are trying to say here: : : Also, how can the authors say the area is only 3.4m2, given the description earlier? This part feels rather confusing.

**Response 8:**

P6\_L2-6 and L12-16 version 1 manuscript. We delete these parts and try to focus on the damage by tsunami as you suggesting.

P6\_L24 version 1 manuscript. We revise its area = 244.7 m2, thanks for the precise comment. We measured size of bridge which moved from original position with intention to give data about bridge dimension that may be used by modeler to assess tsunami force (drag force, lift force, etc.). We provide sketch of the bridge and put it on supplement.

**Change in manuscript 8:**

P9 L8-10 marked-up/revised manuscript

..... Based on these dimensions, the surface area of the bridge was 244.7 m2, the volume was 23.4 m3, and the mass was estimated to be around 56 tons. .....

**Comment 9:**

P7 L2. From which sites? What is the point the authors are trying to make here? P7 L4-10. What is the point of this talk of aftershocks? I suggest all this is deleted, and the authors focus just on the tsunami damage.

**Response 9:**

P7\_L2 version 1 manuscript. We mean it from our measuring sites, Palu bay area. Additional data were documented beside runup and inundation measurement. But, we deleted it. P7\_L4-10 version 1 manuscript. We deleted section about aftershock in order to be more focused. Thanks.

**Change in manuscript 9:**

Please see marked-up/revised manuscript.

**Comment 10:**

Same for P8 L10-19 P8 L20-30 What is the point the authors are making here? The authors don't seem to conclude anything, and merely state conjecture. This might be ok if it was in the discussion section of the paper, but this is not it.

**Response 10:**

P8\_L10-19 and P8 L20-30 version 1 manuscript. It has been removed to make more focused.

**Change in manuscript 10:**

Please see marked-up/revised manuscript.

**Comment 11:**

P9 L1-5 version 1 manuscript. What is the point in a scientific paper of stating that surveys are being carried out? The authors should provide details or analysis, or let others do so. Reporting thatsomething is happening is journalistic.

P9 L19-22 version 1 manuscript. There are already papers that are describing the location of landslides. Also, it is strange that the authors conclude this when they did not talk about this at length in their own paper (they should focus on the conclusions that can be derived from their own work).

**Response 11:**

P9\_L1-5 version 1 manuscript. Thanks for the advice. We deleted "Therefore, the Indonesian Navy deployed the KRI Spica Ship, ..... to conduct a bathymetry survey of Palu Bay after the tsunami.".

P9\_L19-22 version 1 manuscript. We modified this part "Land subsidence in Donggala City (10,068 m2) and Lero Village (22,971 m2) gives evidence for underwater landslides. However, it does not mean there were only two underwater landslides; more landslide locations may be found in the disaster area. " to be "Coastal landslides detected by our team in Donggala City (lost surface area of 10,068 m2) and Lero Village (lost suface area of 22,971 m2) gives additional evidence towards coastal landslides found by other team as reported by Arikawa et al. (2018) and Omira et al. (2019)."

**Change in manuscript 11:**

P10 L12-14 marked-up/revised manuscript.

..... The coastal landslides detected by our team in Donggala City (lost surface area of 10,068 m2) and Lero Village (lost suface area of 22,971 m2) are additional evidence of the coastal landslides found by other teams. .....

**MINOR COMMENTS**

**Comment 12:**

P2 L26 "Most of the victims came from": : :

P2 L30 astonished should not be used in academic literature.

P2 L30, by now you said many times that the earthquake took place due to an activestrike-slip fault in Indonesia. Please delete

P2 L32, again, you repeated many times that the earthquake destroyed many buildings.

**Response 12:**

P2\_L26 version 1 manuscript. We revised it to be"Most of the victims came from..."

P2\_L30 version 1 manuscript. We replaced "astonished" with "surprised"

P2\_L30 version 1 manuscript. We removed "The Palu-Koro fault which divides Sulawesi into two parts, has quite active tectonic activity...."; Moved "the movement of ..."; and "This is the second most active fault in Indonesia after the Yapen fault inPapua." .....

P2\_L32 version 1 manuscript. We deleted it and reduce repeated words or sentences.

**Changes in manuscript 12:**

P2 L20 marked-up/revised manuscript

..... Most of the victims came from this city. .....

P2 L24 marked-up/revised manuscript

•••••

This disaster in Central Sulawesi surprised the scientific community. For a strike-slip fault, the plates move horizontally and thus do not usually cause enough vertical deformation to trigger a huge tsunami. It is still uncertain whether the tsunami was caused by co-seismic deformation or non-tectonic sources. Ulrich et al. (2019) believe that a source related to earthquake displacements is probable and that landsliding may not have been the primary source of the tsunami. In contrast, Takagi et al. (2019), Sassa and Takagawa (2018), and Arikawa et al. (2018) believe that landslides produced the tsunami. Field surveys are important for determining the actual cause.

•••••

P2 L6 marked-up/revised manuscript

..... The movement of rock formations is 35-44 mm/year (Bellier et al., 2001). .....

**Comment 13:**

P3 L11 "for a numerical model"

P3 L12 "rebuilding of the affected areas by the 2018: : :"

P3 L30 what is the point of saying that the authors took videos. Are these provided in the present research or any additional information? Otherwise delete: : :

**Response 13:**

P3\_L11 version 1 manuscript. OK. We change it.

P3\_L12 version 1 manuscript. OK. We change "..... rebuilding the affected areas of the 2018 Sulawesi Tsunami" with "rebuilding of the affected areas by the 2018..."

P3\_L30 version 1 manuscript. We deleted "We also recorded videos and took photographs."

**Changes inmanuscript 13:**

P2 L35 - P3 L1-2 marked-up/revised manuscript

..... For instance, Lynett et.al (2003) employed the field survey data of the 1998 Papua New Guinea tsunami as validation for numerical models, namely the Boussinesq model and a nonlinear shallow water wave model. .....

P3 L4-5 marked-up/revised manuscript

..... More broadly, these data can be used for disaster mitigation and rebuilding of the affected areas by the 2018 Sulawesi Tsunami....

**P4 L3-4 marked-up/revised manuscript**

..... Therefore, our team searched for video recordings and photographs made by local residents while conducting the measurement survey.

**Comment 14:**

P4 L1 you repeated already that this road runs parallel to the coastline.

P4 L9-10, delete these lines.

P4 L17 "until the date of the end of the survey: : :"

P4 L28 "The authors obtained important information from the surveys": : :

P4 L31 "The first wave acted as a trigger for evacuation, with many people starting to escape from the coastline". The technical word is "trigger". Please read other papers about evacuation triggers for tsunamis

**Response 14:**

P4\_L1 version 1 manuscript. We reduced the repeated phrase "parallel to the coastline" and now only one.

P4\_L9-10 version 1 manuscript. We deleted these lines "Many locations with steep cliffs and tsunami trails were not easily visible. We did not take measurements in such locations. Likewise, we did not measure places not significantly affected by the tsunamis."

P4\_L17 version 1 manuscript. We modified "Since the earthquake incident until the date of our team's return " to be "Since the earthquake incident until the date of the end of the survey" P4\_L28 version 1 manuscript. We modified "We got some important information from the interviews to be "The authors obtained important information from the surveys"

P4\_L31 version 1 manuscript. We modified "The first wave was a warning so many people went away from the coastline immediately" to be "The first wave acted as a trigger for evacuation, with many people starting to escape from the coastline".

**Changes in manuscript 14:**

P4 L6-7 marked-up/revised manuscript

..... The road connecting the provinces on Sulawesi island, called the Trans Sulawesi Road, is mostly parallel to the coastline of the bay. .....

**P4 L20-21 marked-up/revised manuscript**

..... From the earthquake incident until the end of the survey, it rained four times, three of which occurred during our survey period, with a duration of less than 2 hours and with low to moderate intensity. .....

P4 L30 marked-up/revised manuscript

..... The authors obtained important information from the surveys, ... .....

**P4 L33-34 marked-up/revised manuscript**

..... The first wave acted as a trigger for evacuation, with many people escaping the coastline. Without this first low-amplitude wave, there may have been more casualties. .....

**Comment 15:**

P5 L5 "is recorded at the maximum horizontal inundation distance". (Delete "the horizontal distance flooded by the wave)

P5 L10 what do the authors mean by tsunami border?

P5 L 12 Delete sentence starting by "The items scattered"

P5 L17 what is "tsunami creeping"? run-up?

P5 L23 "This area was flattened by the tsunami (Fig 6a), with no buildings surviving"?

P5 L26 Rephrase "being quite nimble" P5L27 what exactly is this important observation? Be specific. P5 L32 tsunami risk managers know they can use this data for run-up modelling. Please delete this sentence, it is obvious.

**Response 15:**

P5\_L5 version 1 manuscript. Done. We deleted "the horizontal distance flooded by the wave" P5\_L10 version 1 manuscript. It is meant limit of inundation. We deleted "from coastline to tsunami border on land", and made new sentences.

P5\_L12 version 1 manuscript. Done. We delete the sentence and made new sentences in Part 5 about damage observation.

P5\_L17 version 1 manuscript. Right. We mean "creeping" was runup. We replaced "creeping" by "runup".

P5\_L23 version 1 manuscript. We deleted "This area was flattened by tsunami (Fig. 6a), buildings collapse." And added "caused a few building surviving" in Part 5 about damages.

P5\_L26 version 1 manuscript. We replaced "being quite nimble" by "have agility to save ..."

P5\_L27 version 1 manuscript. We deleted "This is an important observation for future mitigation efforts."

**Changes in manuscript 15:**

P5 L14-15 marked-up/revised manuscript

..... In the simplest case, the runup value is recorded at maximum horizontal inundation distance (IOC Manuals and Guides No. 37, 2014). .....

**P6 L5-6 marked-up/revised manuscript**

.....

North of Tondo is Site 11 (Layana). The topography of this site is relatively flat with a slope of 0.013 (1.3%). Because of this sloping topography, the tsunami wave reached as far as 488 m inland. .....

**P8 L20-22 marked-up/revised manuscript**

..... This site is a trade complex that supports the economic activities of Palu City in particular and Central Sulawesi Province in general. The buildings damaged at this site functioned as shops, warehouses, and corporate offices.

**P6 L1-2 marked-up/revised manuscript**

..... The runup height of 10.73 m is the highest in this survey (Fig. 5) caused a few building surviving.

**P8 L18-19 marked-up/revised manuscript**

..... This is likely due to most of the young residents having the agility to save themselves when the tsunami arrived.

**Comment 16:**

Same for P6 L31-32. P6 L18 check reference, not shown P8 L23 the words "impacted areas with a relatively narrow width" are unclear. Revise. P8 L28 "Ulrih et al. (2019) assume that a: : :"

**Response 16:**

P6 L31-32 version 1 manuscript. We deleted "This phenomenon can be further analyzed to determine the tsunami force."

P6 L18 version 1 manuscript. OK, it refers to Fig 6b.

P8\_L23 version 1 manuscript. We removed part 5.5 about underwater landslides so that we deleted paragraph contained "This suspected source should be located near the impacted areas with a relatively narrow width (Muhari et al., 2018).".

P8\_L28 version 1 manuscript. We revised and move this part "Ulrich et al. (2019) assume that a ..." to introduction part.

**Changes in manuscript 16:**

P9 L15-16 marked-up/revised manuscript

..... The position of this bridge is at the end of Palu Bay (-0.88123°S 119.83907°E). .....

P9 L4-6 marked-up/revised manuscript

Detail measurements were taken of a reinforced concrete bridge with simple support beams on Cumi-cumi Road, Palu City (Fig. 6(b)). This bridge shifted by as far as 9.7 m. It provided clues regarding the strength of the tsunamis. ....

**P2 L26-27 marked-up/revised manuscript**

..... Ulrich et al. (2019) believe that a source related to earthquake displacements is probable and that landsliding may not have been the primary source of the tsunami. .....

**Referee 2 – Prof. Ahmet Cevdet Yalciner**

We would like to thank Prof Ahmet Cevdet Yalciner for the constructive comments and suggestions towards improving our manuscript. We summarize comments from Referee 2, author's response, and author's changes in manuscript. At the end of this authors response, we provide a manuscript showing highlights to mark referee comments. Then follow by a marked-up manuscript showing highlight to mark modification by author. Final form without highlight is available as revised manuscript.

**Comment 1:**

Page-1

Line 14: Indicating the name of the university mentioned would provide more clear information.

Line 22-23: Do the authors have any reference for the earthquake parameters given?

Line 23-24: Do the authors have any reference for the numbers reported?

**Response 1:**

Page-1

- P1 L14 version 1 manuscript. The name of the university is Tadulako University. We move it from abstract to paragraph in section 5 page 8 line 13-14.
- P1 L22-23 version 1 manuscript. The earthquake parameters given are from Meteorological, Climatological and Geophysical Agency (Indonesian: Badan Meteorologi, Klimatologi, dan Geofisika, BMKG) (http://inatews.bmkg.go.id/?act=tsuevents and https://www.bmkg.go.id/press-release/?p=gempabumi-tektonik-m7-7-kabupaten-donggala- sulawesi-tengah-pada-hari-jumat-28-september-2018-berpotensi-tsunami&tag=pressrelease&lang=ID ). The parameters are also available in BMKG's earthquake catalog.
- P1 L23-24 version 1 manuscript. The numbers reported are from the Indonesian National Disaster Management Agency (Indonesian: Badan Nasional Penanggulangan Bencana, BNPB) which was broadcasted by media. We took one of them from https://nasional.tempo.co/read/1138400/jumlah-korban-tewas-terkini-gempa-dan-tsunamipalu-2-113-orang/full&view=ok

We add these references in the manuscript.

**Changes in manuscript:**

P8 L13-14 marked-up/revised manuscript.

- "... This area has many private boarding houses for students of the University of Tadulako, the biggest university in the city of Palu." ...
- P1 L19-21 marked-up/revised manuscript
- On Friday, September 28, 2018, at 18:02:44 Central Indonesia Time (UTC + 8), Palu Bay was hit by a strong earthquake with magnitude  $M_w = 7.5$ . The epicenter was located at -0.22°N 119.85°E at a depth of 10 km and 27 km northeast of Donggala City (BMKG, 2018). .....

**P1 L21-22 marked-up/revised manuscript**

..... As of October 21, 2018, as many as 2,113 people were killed, 1,309 missing, and 4,612 injured (Hadi and Kurniawati, 2018). .....

**Comment 2:**

Page-3

- Line 2-5: "For tsunamis, post-incident surveys are often carried out. Major tsunamis such as: : :." Only stating some of the tsunami post-event surveys like "giving examples" may not be appropriate. Two suggestions: Either state the importance and relation of them with this study OR delete these sentences.
- Line 9: "Observation of damage was also conducted." Too general sentence. What kind of damage data is collected? Any details on the data collection processes?

**Response 2:**

- P3 L2-5 version 1 manuscript. We delete these sentences "For tsunamis, post-incident surveys are often carried out. Major tsunamis such as: : :."
- P3 L9 version 1 manuscript. We emphasized damage to buildings and infrastructures. We identified the difference about damage by earthquake, liquefaction and tsunami. We made videos to document damage along Trans Sulawesi Road, compared them to Google Street View®, Google Map and Google Earth and concluded that the severe damage was limited in 150 m from coastline. We also measured dimension of a bridge because we assume it was special case we found. We add about this in paragraphs of the manuscript.

**Changes in manuscript 2:**

**P2 L31-31 marked-up/revised manuscript**

..... Tsunami flow depth on land was also measured in some sites. In addition, tsunami arrival time was analyzed and observation of buildings and infrastructures damage was conducted. .....

**P5 L8-12 marked-up/revised manuscript**

•••••

Damage observation was carried out at each site of surveys. We emphasized on damage to buildings and infrastructures although other kind of damage are interesting, such as vegetation, shoreline, and properties (cars, boats, fisherman tools, etc.). Videos and photographs were produced to assess the damage. Video recorded along trans Sulawesi Road were compared to Google Street View, Google Map, and Google Earth in order to assess the distance of damage from coastine. In addition, detail measurement of dimension was done for special object (for instance bridge) which is useful for tsunami force analysis.

**Comment 3:**

Page 9:

- Line 14: "There were three main tsunami waves that reached the beach." Which beach? Not clear.
- Line 14: "The first wave was relatively low." With respect to what? You should State it more clearly and in an understandable way.
- Line 18: "The wave that hit the beach was quite high." This sentence by itself does not provide any meaningful information.

**Response 3:**

- P9 L14 version 1 manuscript. We modify "There were three main tsunami waves that reached the beach." to be "Most people interviewed in the survey area testified that there were three main tsunami waves that reached the coastal zone in Palu Bay. The second was the highest.".
- P9 L18 version 1 manuscript. We replace "The wave that hit the beach was quite high" with "The tsunami waves that hit the coastal zone in Palu Bay were very strong, as indicated by massive damage at each site we surveyed. Severe damage was limited to within 150 m from the coastline. These include the shifting of a 56-ton bridge."

**Changes in manuscript 3:**

**P10 L5-7 marked-up/revised manuscript**

..... Most people interviewed in the survey area testified that there were three main tsunami waves that reached the coastal zone in Palu Bay. The second was the highest. .....

**P10 L1-2 marked-up/revised manuscript**

..... The tsunami waves that hit the coastal zone in Palu Bay were very strong, as indicated by massive damage at each site we surveyed. Severe damage was limited to within 150 m from the coastline. These include the shifting of a 56-ton bridge. .....

**Comment 4:**

General Comments: - A brief summary and citation of previously published papers on 2018 Palu Event field survey is necessary. - Section 5.1, Aftershock information is not related with the focus of this study and the work done. - The conclusion section should be rewritten by clear sentences and providing a comprehensive summary of the results obtained. For example, Giving ranges such as "2 to10 m and the inundation distances were 80 to 500 m." Or "The arrival time of waves varied from 3 to 10 minutes.." does not provide satisfying information. The authors, at least, may add the locations of these measurements.

**Response 4:**

- Thanks for the suggestions. We provide a brief summary and citation of previously published papers e.g. Omira et al (2019), Mikami et al (2019), Yalciner et al. (2018), Muhari et al (2018), and Arikawa et al. (2018).
- We remove section 5.1 about aftershock information which is also suggested by Referee 1.

•••••

We modify the sentences, the ranges of numbers are replaced by the maximum and significant run up height and inundation distance with mentioning the locations.

Your comments and suggestions are accomodated in our revised manuscript.

**Changes in manuscript 4:**

**P3 L6-15 marked-up/revised manuscript**

[revised manuscript text omitted]